# Analytic thermodynamic properties of the Lieb-Liniger gas

**Matthew L. Kerr[1,2], Giulia De Rosi[3⋆] and Karen V. Kheruntsyan[2⋆]**

**1** The Rudolf Peierls Centre for Theoretical Physics, Oxford University, Oxford OX1 3NP, UK
**2** School of Mathematics and Physics, University of Queensland,
Brisbane, Queensland 4072, Australia
**3** Departament de Física, Universitat Politècnica de Catalunya,
Campus Nord B4-B5, 08034 Barcelona, Spain

⋆ giulia.de.rosi@upc.edu , † karen.kheruntsyan@uq.edu.au

## Abstract

We present a comprehensive review on the state-of-the-art of the approximate analytic approaches describing the finite-temperature thermodynamic quantities of the Lieb-Liniger model of the one-dimensional (1D) Bose gas with contact repulsive interactions. This paradigmatic model of quantum many-body-theory plays an important role in many areas of physics—thanks to its integrability and possible experimental realization using, e.g., ensembles of ultracold bosonic atoms confined to quasi-1D geometries. The thermodynamics of the uniform Lieb-Liniger gas can be obtained numerically using the exact thermal Bethe ansatz (TBA) method, first derived in 1969 by Yang and Yang. However, the TBA numerical calculations do not allow for the in-depth understanding of the underlying physical mechanisms that govern the thermodynamic behavior of the Lieb-Liniger gas at finite temperature. Our work is then motivated by the insights that emerge naturally from the transparency of closed-form analytic results, which are derived here in six different regimes of the gas and which exhibit an excellent agreement with the TBA numerics. Our findings can be further adopted for characterising the equilibrium properties of inhomogeneous (e.g., harmonically trapped) 1D Bose gases within the local density approximation and for the development of improved hydrodynamic theories, allowing for the calculation of breathing mode frequencies which depend on the underlying thermodynamic equation of state. Our analytic approaches can be applied to other systems including impurities in a quantum bath, liquid helium-4, and ultracold Bose gas mixtures.

# 1  Introduction

The investigation of complex quantum many-body systems such as, for example, ensembles of interacting atoms or electrons, is a forefront research topic at the interface of chemistry, materials science, solid-state and condensed matter physics at low temperatures. The Lieb-Liniger model [1, 2] is a versatile testbed for such studies; it describes a system of identical bosons interacting with a contact pairwise repulsive interaction, confined to one spatial dimension (1D) [3–5]. Interestingly, at very strong interactions, the thermodynamic properties of this bosonic system approach those expected of an ensemble of ideal (noninteracting) fermions [6].

The Lieb-Liniger model may be precisely simulated by cavity quantum-electrodynamic devices [7]. It has been also experimentally realized in superconducting circuits crossing weak to strong repulsions [8] and in optical fibres with Kerr nonlinearities [9]. However, the best experimental platform for its realization is provided by ultracold quantum gases [10,11]. Indeed, in quantum gas experiments, samples of up to $\sim 10^6$ atoms can be confined to 1D geometry in highly anisotropic trapping potentials typically realized using atom chips [11–13] or two-dimensional (2D) optical lattices [14–18]; furthermore, these gases are typically so dilute and cold that the complex interatomic interactions are dominated by the relatively simple elastic two-body processes in the (low-energy) $s$-wave scattering channel [19]. This in turn provides a nearly ideal realization of the Lieb-Liniger model, with the added benefit that the interatomic interaction strength can be tuned at will, via either a magnetic Feshbach resonance [20] or a confinement induced resonance [19, 21, 22].

Apart from being realisable in ultracold atom experiments, the Lieb-Liniger gas constitutes a paradigmatic example of a quantum *integrable* (or *exactly solvable*) model of an interacting many-particle system [23, 24]. Exploiting the rich mathematical structure of integrable mod-

els allows for the direct comparison between exact and analytical results, hence offering deep insights and understanding into underlying physics that are not often available within generic (nonintegrable) quantum systems. At zero temperature, the ground state energy and the excitation spectrum of the Lieb-Liniger gas can be calculated exactly [1,2] by means of the Bethe ansatz method [25,26]. At finite temperature, by using the exact integrability of the Lieb-Liniger model, the thermodynamic properties of the 1D Bose gas can be obtained numerically via the celebrated thermal or thermodynamic Bethe ansatz (TBA) approach within the Yang-Yang thermodynamic theory [27,28]. While the TBA method provides exact results for various thermodynamic quantities by benchmarking the corresponding analytical predictions and experimental measurements, its numerical implementation is often computationally demanding, especially at sufficiently low or high temperatures. The development of simple analytical theories, as is done in this work, offers instead a more transparent and computationally efficient framework for exploring the thermodynamics of 1D Bose gases.

The Lieb-Liniger gas is characterised by an extremely rich landscape of different physical regimes [29–32], which can be explored by changing the interaction strength and temperature. These regimes are all separated by smooth crossovers (rather than by critical phase transitions) and they might or might not share the typical features of superfluids and three-dimensional (3D) Bose-Einstein condensates. Thermodynamically distinct regimes can be classified by the behaviour of the local atom-atom correlation function $g^{(2)}(0)$ [30,31], which itself is a thermodynamic quantity that can be calculated exactly using the TBA. However, in most cases, the full correlation function $g^{(2)}(r)$ (as a function of the relative distance $r$ between two atoms) and other various correlation properties cannot be evaluated using TBA and require instead more advanced *ab-initio* techniques to explore a wide range of interaction strengths and temperatures [33–54], which cannot be accessed with approximate analytical limits. In addition, such correlation functions are often hard to measure experimentally.[1]

The different regimes in a 1D Bose gas can alternatively be identified with the corresponding characteristic behavior of other thermodynamic quantities, which are easier to measure in ultracold atom experiments [55–58]. In the past years, considerable attention has been dedicated to such asymptotic analytical limits [30–32,40,59–64] and their comparison with the exact solution provided by the TBA approach. Weakly-interacting 1D Bose systems at sufficiently low temperatures form a phase-fluctuating quasicondensate and can be well-described via various mean-field methods [65] and Bogoliubov theory [34,63,64]. Conversely, in the limit of large repulsive interaction strengths, the thermodynamic behavior of the Lieb-Liniger model resembles that of an hard-core system [63,64] which itself is similar to the ideal Fermi gas. For very high temperatures, thermal fluctuations dominate over quantum correlations and the system exhibits the classical gas behavior [63,64]. Despite these results, the thermodynamic properties of a weakly-interacting 1D Bose gas at intermediate temperatures have not been fully explored. In particular, this regime poses unique challenges due to the complicated interplay of quantum and thermal effects which defy easy physical interpretation.

Very recently, it has been understood that the 1D Bose gas for arbitrary contact interaction strength exhibits the *hole anomaly*, i.e., a thermal feature in the thermodynamic properties as a function of temperature, identified by a peak in the specific heat or a maximum in the chemical potential, located at the anomaly temperature [64]. The anomaly mechanism is due to the thermal occupation of unpopulated states located below the hole branch in the excitation spectrum. The maximum of the hole branch provides the energy scale for the anomaly temperature $T_A$. When the temperature of the gas $T$ is comparable to $T_A$, empty spectral

---

[1]The only experimental measurement of the local atom-atom correlation $g^{(2)}(0)$ in a 1D Bose gas was carried out back in 2005 [18]. Here it was measured at very low temperatures, for a range of interaction strengths in an array of 1D Bose gas tubes formed by a 2D optical lattice, implying that the measured signal was an average over different 1D Bose gases in different physical regimes.

states become thermally occupied so that the excitations experience the breakdown of the low-temperature quasiparticle description, and thermal fluctuations dominate over quantum correlations at temperatures higher than the anomaly threshold $T > T_A$. Thus, the internal energy is almost constant with temperature at $T < T_A$, and rapidly increases at $T > T_A$ [54].

In the present work, we derive simple analytically tractable limits for the thermodynamic quantities of the uniform 1D Bose gas. We explore all possible regimes (identified in Refs. [30, 31]) crossing from weak to strong interactions and from low to high temperatures. In doing so, we obtain the complete description for the thermal quasicondensate regime that has not been previously reported. In addition, we find new interaction and temperature dependent terms for the degenerate and non-degenerate nearly ideal Bose gas regimes. We demonstrate that our analytic results are in excellent agreement with the well-established exact TBA solution in their validity ranges of values of interaction and temperature parameters and far away from the hole anomaly. Moreover, we revise and refine the conditions defining the crossover boundaries between different regimes of the gas. Finally, we show that our analytic findings can be generalized to inhomogeneous systems where the local density approximation is valid. To this aim, we construct the density profiles of a harmonically trapped 1D Bose gas in different finite-temperature regimes, which go beyond the previous, widely used approximations (such as the Thomas-Fermi inverted parabola) and we compare them with the TBA predictions. We show how these density profiles, particularly their fits to the thermal tails of the trapped 1D Bose gases, can be used for the experimental extraction of temperature (thermometry).

The structure of the paper is as follows. In Sec. 2 we introduce the Lieb-Liniger model, describing a uniform 1D Bose gas at finite temperature, and the Yang-Yang theory for the exact calculation of the thermodynamic properties. Sec. 3 is devoted to a review of the different asymptotic regimes emerging in the system and relying on the behavior of the local pair correlation function. In Sec. 4, we report the analytical approximations for several thermodynamic properties in each of the respective regimes. In addition, we provide the generalization of our analytical results to harmonically trapped configurations. In Sec. 5, we demonstrate the validity of our analytical limits for the local pair correlation function, the Helmholtz free energy and the chemical potential in the uniform configuration, by comparison with the exact TBA solution. Finally, in Sec. 6, we draw the conclusions, the possible upcoming experimental observations, and perspectives of our work.

## 2 Model

Consider a uniform ensemble of $N$ bosons interacting via a two-body contact potential in a 1D box of length $L$ and with periodic boundary conditions. The corresponding Lieb-Liniger (LL) Hamiltonian is given by [1]

$$H_{\mathrm{LL}} = -\frac{\hbar^2}{2m} \sum_{i=1}^{N} \frac{\partial^2}{\partial x_i^2} + g \sum_{i<j} \delta(x_i - x_j), \tag{1}$$

where $m$ is the bosonic particle mass and $g$ is the 1D interaction strength (coupling constant) related to the 1D $s$-wave scattering length $a_{1\mathrm{D}}$ via $g = -2\hbar^2/(ma_{1\mathrm{D}})$ [19]; we assume here that the interactions are repulsive, which corresponds to $g > 0$ (or $a_{1\mathrm{D}} < 0$). The LL model is integrable and exactly solvable using the Bethe ansatz method [1, 2]. In second quantized form, the above Hamiltonian can be rewritten as

$$\hat{H}_{\mathrm{LL}} = -\frac{\hbar^2}{2m} \int_0^L dx\, \hat{\psi}^\dagger(x) \frac{\partial^2}{\partial x^2} \hat{\psi}(x) + \frac{g}{2} \int_0^L dx\, \hat{\psi}^\dagger(x)\hat{\psi}^\dagger(x)\hat{\psi}(x)\hat{\psi}(x), \tag{2}$$

where $\hat{\psi}(x)$ is the bosonic field operator.

The LL model can be experimentally realised by confining ultracold atomic gases to highly anisotropic confining potentials, which, for simplicity, are assumed to be cylindrically symmetric and harmonic in the transverse (radial) and longitudinal (axial) directions. The necessary condition for achieving the 1D geometry is that the frequency of the transverse confining trap (along the $y$ and $z$ axes), $\omega_\perp = \omega_y = \omega_z$, is much larger than the frequency of the longitudinal ($x$ axis) trap, $\omega_\perp \gg \omega_x$. Additionally, one must ensure that all relevant energy scales in the problem, such as the chemical potential $\mu$ and the average thermal energy $k_B T$, are much smaller than the energy of the first transversely excited state, $\{\mu, k_B T\} \ll \hbar\omega_\perp$ [66]. Under such 1D conditions, the radial harmonic trap is strong enough such that the transverse excitations will be negligible, implying that the dynamics of atoms is restricted only to the longitudinal direction, whereas it is effectively frozen in the transverse directions. In this case, the coupling constant $g$ in the LL Hamiltonian (1) can be written down as $g \simeq 2\hbar\omega_\perp a_{3D}$, i.e., in terms of the three-dimensional (3D) $s$-wave scattering length $a_{3D} > 0$, where we have additionally assumed that the value for $a_{3D}$ is far away from a confinement induced resonance [19, 29].

The same considerations apply if the longitudinal confinement is not harmonic, in which case the highly anisotropic nature of the confining potential can be reformulated in terms of the characteristic lengthscale in the axial direction, $l_x$, and the harmonic oscillator length in the transverse direction, $a_\perp^{(\text{osc})} = \sqrt{\hbar/(m\omega_\perp)}$, requiring that $l_x \gg a_\perp^{(\text{osc})}$. More generally, if the atomic ensemble is sufficiently large, the boundary effects can be safely neglected, so that the LL Hamiltonian is applicable to systems that are not necessarily periodic or even uniform. In the latter case, the spatial inhomogeneities due to the nonuniform longitudinal confining potential $V(x)$ can often be dealt with using the local density approximation [31, 67].

The strength of interactions between particles in a uniform 1D Bose gas is encoded by the dimensionless parameter

$$\gamma = \frac{mg}{\hbar^2 n} = -\frac{2}{n a_{1D}}, \tag{3}$$

where $n = N/L$ is the 1D (linear) density. When $\gamma \ll 1$, the system is weakly interacting. Conversely, for $\gamma \gg 1$, the system is in the strongly interacting regime, approaching that of impenetrable or hard-core bosons in the limit $\gamma \to \infty$ [1, 6]. One can equivalently enter the strongly-interacting regime $\gamma \gg 1$ by either increasing the coupling constant $g$ or *decreasing* (rather than *increasing*) the 1D particle number density $n$, with the latter option being a counter-intuitive result not encountered in 3D systems.

At non-zero temperatures, one needs a second parameter—a dimensionless temperature $\tau$ —to characterise the LL model. This can be introduced by scaling the temperature $T$ of the system by the temperature of quantum degeneracy of the 1D Bose gas, $T_d = \hbar^2 n^2 / (2 m k_B)$,

$$\tau = \frac{T}{T_d} = \frac{2 m k_B T}{\hbar^2 n^2}. \tag{4}$$

When the quantum degeneracy is reached, $T \sim T_d$ or $\tau \sim 1$, the thermal de Broglie wavelength associated to each atom, $\lambda_T = \sqrt{2\pi\hbar^2/(m k_B T)}$, is of the order of the mean interparticle distance $1/n$. This is the temperature threshold below which quantum effects begin to dominate.

The two dimensionless parameters, $\gamma$ and $\tau$, completely characterise the thermodynamic properties of a uniform 1D Bose gas. Such thermodynamic theory has been indeed constructed in 1969 by C. N. Yang and C. P. Yang [27], who extended the work of Lieb and Liniger [1] to finite temperatures. Yang and Yang also proved the analyticity of thermodynamic quantities, indicating the absence of any phase transition for all temperatures. Their approach established the quasi-particle formulation of the thermal Bethe ansatz method, which we outline here.

As demonstrated by Lieb and Liniger, the quantum many-body eigenstates of the Hamiltonian, Eq. (1), are characterised by a set of distinct quantum numbers corresponding to a set of

quasi-momenta $k$ via the Bethe ansatz equation [1]. In the thermodynamic limit ($N, L \to \infty$, with $n = N/L$ fixed), one can then consider the distribution of these quasi-momenta of particles, $\rho_p(k)$, and of holes $\rho_h(k)$, where the latter refers to the quasi-momenta of unoccupied states from the entire set of possible quasi-momenta. These two distributions are related via the integral equation [27],

$$2\pi\left(\rho_p(k) + \rho_h(k)\right) = 1 + \frac{2mg}{\hbar^2} \int_{-\infty}^{\infty} \frac{\rho_p(k')\,dk'}{(mg/\hbar^2)^2 + (k-k')^2}.\tag{5}$$

In terms of the quasi-momentum distributions, one can express the particle density,

$$n = \int_{-\infty}^{\infty} dk\,\rho_p(k),\tag{6}$$

the total internal energy,

$$E = L \int_{-\infty}^{\infty} dk\,\rho_p(k)\frac{\hbar^2 k^2}{2m},\tag{7}$$

and the entropy of the system,

$$S = k_B L \int_{-\infty}^{\infty} dk\left[(\rho_p + \rho_h)\ln\left(\rho_p + \rho_h\right) - \rho_p \ln\left(\rho_p\right) - \rho_h \ln\left(\rho_h\right)\right].\tag{8}$$

While Eq. (5) uniquely determines the distribution of holes $\rho_h(k)$, given that of particles $\rho_p(k)$, the equilibrium quasi-momentum distribution, $\rho_p(k)$, itself is obtained from

$$\frac{\rho_h(k)}{\rho_p(k)} = e^{\varepsilon(k)/(k_B T)},\tag{9}$$

where the excitation spectrum $\varepsilon(k)$ is the solution to the integral equation,

$$\varepsilon(k) = \frac{\hbar^2 k^2}{2m} - \mu - \frac{k_B T}{\pi} \int_{-\infty}^{\infty} \frac{(mg/\hbar^2)\,dk'}{(mg/\hbar^2)^2 + (k-k')^2} \ln\left(1 - e^{-\varepsilon(k')/(k_B T)}\right),\tag{10}$$

which can be obtained by minimising the Helmholtz free energy functional $F = E - TS$ with respect to $\rho_p(k)$. Here, $\mu$ is the chemical potential which ultimately determines the total number of particles $N$ in the system, and the integral equation itself can be solved for $\varepsilon(k)$ iteratively [27].

With the use of Eq. (9), the left-hand side of Eq. (5) can be rewritten as $2\pi\rho_p(k)\left(1 + \exp[\varepsilon(k)/(k_B T)]\right)$, and the resulting integral equation can then be solved for $\rho_p(k)$ iteratively [27]. The solution for $\rho_p(k)$ then allows one to calculate the particle number density $n$, the total internal energy $E$, and the entropy $S$, using Eqs. (6), (7), and (8), respectively. Finally, given $\varepsilon(k)$, the pressure of the system can then be obtained via [27]

$$P = \frac{k_B T}{2\pi} \int_{-\infty}^{\infty} dk \ln\left(1 + e^{-\varepsilon(k)/(k_B T)}\right),\tag{11}$$

which itself allows one to calculate the Helmholtz free energy,

$$F = -PL + \mu N,\tag{12}$$

and hence all other thermodynamic quantities of the system.

# 3  Regimes of the uniform 1D Bose gas in thermal equilibrium

The Lieb-Liniger model (1) possesses six asymptotic regimes, as first identified by Kheruntsyan *et al.* in 2003, on the basis of the form of the normalised local (same point) two-atom or pair correlation function [30] which describes the probability of two particles to overlap [65]:

$$g^{(2)}(0) = \frac{\langle \hat{\psi}^\dagger(x)\hat{\psi}^\dagger(x)\hat{\psi}(x)\hat{\psi}(x) \rangle}{n^2}, \tag{13}$$

where $n(x) = \langle \hat{\psi}^\dagger(x)\hat{\psi}(x) \rangle$ is the linear 1D density and $\langle \cdots \rangle$ denotes the ensemble average at thermal equilibrium. These regimes, shown in Fig. 1, can be broadly identified as corresponding to the nearly ideal Bose gas, quasicondensate, and the strongly-interacting system; furthermore, each of the three main regimes can be further divided into two sub-regimes, as we will explain below. The boundaries between all regimes are smooth crossovers and arise when one compares the typical energy scales in the problem, such as the thermal energy $k_B T$, the scattering energy $mg^2/\hbar^2 = \hbar^2/(m|a_{1D}|^2)$, the mean-field interaction energy $gn$, or the Fermi energy $E_F = \hbar^2 \pi^2 n^2/(2m)$. The crossover boundaries can be also equivalently identified by comparing the relevant length scales [29], such as the thermal de Broglie wavelength $\lambda_T$, the thermal phase coherence length $l_\phi = \hbar^2 n/(mk_B T)$, the healing length $l_h = \sqrt{\hbar^2/(mng)}$, the mean interparticle separation $1/n$ or the Fermi wavelength $\lambda_F = 2\pi/k_F$ ($k_F = \pi n$ being the Fermi wavenumber), and the absolute value of the 1D $s$-wave scattering length $|a_{1D}| = 2\hbar^2/(mg)$.[2]

We now provide a brief review of these regimes based on the calculated local pair correlation function $g^{(2)}(0)$ from Ref. [30]. We note, however, that when specifying the conditions of applicability of each of the analytic expressions for $g^{(2)}(0)$ in each regime and presented in Secs. 3.1–3.3 below, we restore the "numerical factors of the order one" that were ignored in Ref. [30]. From the results of more accurate analytical calculations, to be presented in Sec. 4, we find that the restored numerical factors provide more precise crossover boundaries between the different regimes of the 1D Bose gas when compared to the exact TBA prediction. Thus, even though we quote the 2003 analytical expressions for the $g^{(2)}(0)$ [30] for the brevity of presentation, we update the conditions of their applicability, which now serve as improved regime boundaries shown in Fig. 1. As an example, the conditions corresponding to weak interactions and low temperatures $\tau \ll 2\sqrt{\gamma}$, $\tau/2 \ll \gamma$, and $2\gamma \ll \tau \ll 2\sqrt{\gamma}$, appearing in Eqs. (14)–(16), now include factors of 2, which were omitted in Ref. [30]. Similarly, the regime of quantum degeneracy for infinitely strong interactions ($\gamma \to \infty$) is now delimited not via the crossover boundary $\tau = T/T_d = 1$, but via $\tau = \pi^2$. This is indeed more suitable, as the strongly repulsive 1D Bose gas displays "fermionization", owing to the Bose-Fermi mapping in the Tonks-Girardeau limit ($\gamma \to \infty$) whose thermodynamics is identical to that of the ideal Fermi gas [6, 68]. Therefore, a more accurate definition of the temperature of quantum degeneracy in this case is provided by the Fermi temperature $T_F = \hbar^2 \pi^2 n^2/(2mk_B)$, rather than by $T_d = T_F/\pi^2$. Accordingly, the boundary $T = T_F$ between degenerate and non-degenerate regimes for $\gamma \to \infty$ is defined by $\tau = \pi^2$ in a dimensionless form, rather than by $\tau = 1$. Moreover, we have further updated the boundary $\tau = \pi^2$ with $\tau = \pi^2/(1 + 2/\gamma)^2$ from Refs. [63,64], where we have included the effects of large but finite interaction strength $\gamma \gg 1$ via the "negative excluded volume" corrections, within the hard-core model.

---

[2]The healing length can be generalized along the whole interaction crossover if expressed as a function of the sound velocity at zero temperature $v$: $l_h = \hbar/(mv)$. In the limit of weak interactions $v = \sqrt{gn/m}$ and one recovers $l_h = \sqrt{\hbar^2/(mng)}$. In the strongly-repulsive regime, the sound velocity approaches asymptotically the Fermi velocity $v_F = \hbar \pi n/m$ and then the healing length, the Fermi wavelength and the mean interparticle distance all provide the same length scale of the system, $l_h = 1/k_F \sim \lambda_F \sim 1/n$.

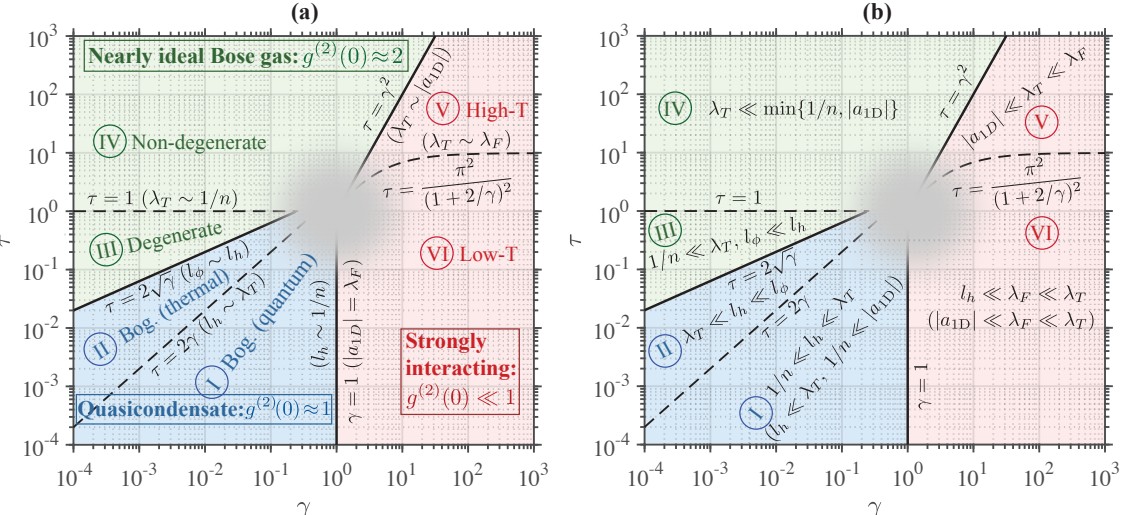

Figure 1: Regimes of the uniform Lieb-Liniger gas at thermal equilibrium in terms of the interaction strength $\gamma$, Eq. (3), and temperature $\tau$, Eq. (4). We distinguish six different thermodynamic regimes, separated by smooth crossovers. The weakly-interacting quasicondensate regime (in blue) is described by the Bogoliubov (Bog.) theory and can be further subdivided into regions I (quantum) and II (thermal). The nearly ideal Bose gas regime (in green) is treated with a perturbative description for small $\gamma$. It includes degenerate (III) and non-degenerate (IV) regions. The strongly-interacting gas (in red) is tackled with the high-temperature (V) and low-temperature (VI) fermionization approaches, respectively. Panel (a) shows the different regimes and their boundaries in terms of $\gamma$ and $\tau$, as well as the characteristic length scales such as the thermal de Broglie wavelength $\lambda_T = \sqrt{2\pi\hbar^2/(mk_BT)}$, the thermal phase coherence length $l_\phi = \hbar^2 n/(mk_BT)$, the healing length $l_h = \sqrt{\hbar^2/(mng)}$, the mean interparticle separation $1/n$ or the Fermi wavelength $\lambda_F = 2/n$, and the absolute value of the 1D $s-$wave scattering length $|a_{1D}| = 2\hbar^2/(mg)$. The blurred grey area around $\gamma \sim 1$ and $\tau \sim 1$ corresponds to the region where no approximate analytic predictions are expected to be valid, which is also true in the vicinity of the crossover boundaries (solid and dashed lines) separating the different regimes. Panel (b) displays the same diagram but spells out the conditions of applicability of our approximate analytic results for the local pair correlation function $g^{(2)}(0)$ and other thermodynamic quantities in terms of inequalities for the characteristic length scales.

## 3.1 Regions I and II: Quasicondensate regime

When the typical internal energy per particle is of the order of the mean interaction energy, and much larger than the scattering energy, $E/N \sim gn \gg mg^2/\hbar^2$, the gas behaves as a quasicondensate emerging only for weak interactions and low temperatures. Physically, the quasicondensate is characterised by suppressed density fluctuations due to the repulsive nature of interactions [12] and, hence, the resulting atom-atom correlations are weak, as approaching the uncorrelated limit of $g^{(2)}(0) \simeq 1$. In contrast to true Bose-Einstein condensates in 3D, the 1D quasicondensate exhibits a fluctuating phase at long wavelengths which is enhanced by thermal excitations [69] and destroys any long-range order behavior [70]. However, the behaviour of the 1D quasicondensate is well-described via Bogoliubov (Bog.) description, or more precisely, the extension of Bogoliubov theory to quasicondensates [29, 34, 71, 72].

In terms of the dimensionless parameters, the quasicondensate regime is identified through the following inequalities in Fig. 1:

$$\gamma \ll 1, \qquad \tau \ll 2\sqrt{\gamma}. \tag{14}$$

The first inequality can be derived immediately from $gn \gg mg^2/\hbar^2$ whilst the second one follows from insisting that density fluctuations are small and hence provide only a small correction to the uncorrelated level of $g^{(2)}(0) = 1$.

By comparing the average thermal energy $k_B T$ to the mean-field interaction energy $gn$, the 1D quasicondensate regime can be further divided into two different regimes. Namely, the quantum quasicondensate regime (region I), dominated by quantum fluctuations ($k_B T \ll gn$ or $\tau \ll 2\gamma$); and the thermal quasicondensate regime (region II), dominated by thermal fluctuations ($k_B T \gg gn$ or $\tau \gg 2\gamma$), see Fig. 1. In terms of the characteristic length scales, the quantum quasicondensate regime corresponds to $1/n \ll l_h \ll \lambda_T$ whilst the thermal quasicondensate regime is defined via $\lambda_T \ll l_h \ll l_\phi$.

The local pair correlation function in the quantum quasicondensate regime I is given by [30, 64],

$$g^{(2)}(0) = 1 - \frac{2}{\pi}\gamma^{1/2} + \frac{\pi}{24}\tau^2\gamma^{-3/2}, \qquad \tau/2 \ll \gamma \ll 1, \tag{15}$$

whilst in the thermal quasicondensate regime II [30],

$$g^{(2)}(0) = 1 + \frac{1}{2}\tau\gamma^{-1/2}, \qquad 2\gamma \ll \tau \ll 2\sqrt{\gamma}. \tag{16}$$

While here in Sec. 3 we are simply quoting the final results for the local pair correlation function from Ref. [30], in Sec. 4 we will re-derive these findings to a higher accuracy, in addition to obtaining other thermodynamic quantities of interest.

## 3.2 Regions III and IV: Nearly ideal Bose gas regime

The nearly ideal Bose gas regime arises at sufficiently high temperatures for which the internal energy per particle far exceeds all the other relevant energy scales, i.e., $E/N \gg \max\{mg^2/\hbar^2, gn\}$. Finite-temperature effects then dominate interactions and both density and phase fluctuations are large. Since interaction effects are negligible, the system effectively behaves as a gas of non-interacting (ideal) bosons. The problem can be then treated using the perturbation theory for small interaction strength $\gamma \to 0$, approaching the ideal Bose gas limit [30, 59]. This regime corresponds to the region in the $(\gamma, \tau)$ parameter diagram of Fig. 1 defined via the inequality [30]

$$\gamma \ll \min\{\tau^2/4, \sqrt{\tau}\}. \tag{17}$$

We stress that the nearly ideal Bose gas regime is not restricted only to the Maxwell-Boltzmann classical limit of very high temperatures, but it also describes a degenerate quantum system at temperatures lower than the temperature of quantum degeneracy, $T < T_d$, corresponding to $\tau < 1$. One can then identify two different regimes. The first one being the degenerate regime (region III in Fig. 1) characterized by the internal energy per particle of the same order as the chemical potential $\mu$ and much smaller than the average thermal energy $k_B T$: $E/N \sim |\mu| \ll k_B T$. The second regime is non-degenerate (region IV) where the internal energy is purely thermal and the magnitude of the chemical potential is much larger than any other relevant energy scales, $E/N \sim k_B T \ll |\mu|$. In the degenerate regime III, Eq.(17) is recovered by the chemical potential expressed in terms of the density [11]

$$|\mu| = \left(\frac{k_B T}{\hbar n}\right)^2 \frac{m}{2}, \tag{18}$$

and by assuming that it is much larger than the chemical potential of the quasicondensate at zero temperature $gn$: $|\mu| \gg gn$. On the other hand, Eq. (17) is obtained in the non-degenerate regime IV from the condition $k_B T \gg mg^2/\hbar^2$. In terms of the characteristic length scales, regime III is identified by the conditions $1/n \ll \lambda_T$ and $l_\phi \ll l_h$, whilst regime IV by $\lambda_T \ll \min\{1/n, |a_{1D}|\}$.

In the degenerate regime (III), the local pair correlation function is given by [30]

$$g^{(2)}(0) = 2 - \frac{4\gamma}{\tau^2}, \qquad 2\sqrt{\gamma} \ll \tau \ll 1. \tag{19}$$

Conversely, by raising the temperature, the system enters into the non-degenerate regime IV which approaches the Maxwell-Boltzmann classical gas limit whilst the local pair correlation function takes the form [30]:

$$g^{(2)}(0) = 2 - \gamma\sqrt{\frac{2\pi}{\tau}}, \qquad \tau \gg \max\{1, \gamma^2\}. \tag{20}$$

The large thermal fluctuations present in regimes III and IV lead to the atom *bunching* effect, for which the probability of observing two atoms in the same position is enhanced (indicated by $g^{(2)}(0) \simeq 2$) compared to uncorrelated atoms with $g^{(2)}(0) = 1$ [30, 40, 53, 59].

## 3.3 Regions V and VI: Strongly interacting regime

The strongly interacting regime is attained when the scattering energy $mg^2/\hbar^2$ is the dominant energy scale in the system, $mg^2/\hbar^2 \gg \max\{E/N, gn\}$. This condition identifies the regions V and VI in the $(\gamma, \tau)$ parameter space of Fig. 1 and which satisfy

$$\gamma \gg \max\{1, \sqrt{\tau}\}. \tag{21}$$

In the Tonks-Girardeau limit [6, 68] of infinite repulsive interaction strength $\gamma \to \infty$, the bosonic atoms become impenetrable and cannot occupy the same position – mimicking an effective Pauli exclusion principle satisfied by non-interacting fermions. This leads to the fermionization of the bosonic particles, whose many-body wavefunction vanishes at zero relative interatomic distance. The established Bose-Fermi mapping [6] justifies the applicability of the ideal Fermi gas model for the description of the thermodynamics in the Tonks-Girardeau limit. On the other hand, the regime of large but finite interaction strengths, $\gamma \gg 1$, can be treated within the perturbation theory for small values of $1/\gamma$, close to the ideal Fermi gas limit, wherein the Bose-Fermi mapping still holds [63, 64, 73, 74].

The strongly-interacting regime can be further distinguished into two regimes appearing in Fig. 1. Namely, the regime of high-temperature fermionization (region V), defined via $\pi^2/(1 + 2/\gamma)^2 \ll \tau \ll \gamma^2$ and the one of quantum degeneracy at low temperatures $\tau \ll \pi^2/(1 + 2/\gamma)^2$ (region VI). The above conditions, expressed in terms of the characteristic length scales of the Lieb-Liniger gas, correspond to $|a_{1D}| \ll \lambda_T \ll \lambda_F$ (regime V) and $l_h \ll \lambda_F \ll \lambda_T$ (regime VI).

In the high-temperature fermionization regime V, the local pair correlation function is given by [30, 64, 75]

$$g^{(2)}(0) = \frac{2\tau}{\gamma^2}, \qquad \pi^2/(1 + 2/\gamma)^2 \ll \tau \ll \gamma^2. \tag{22}$$

Conversely, in the low-temperature fermionization regime VI, one finds [30, 64, 75]:

$$g^{(2)}(0) = \frac{4}{3}\left(\frac{\pi}{\gamma}\right)^2\left[1 + \frac{\tau^2}{4\pi^2}\right], \qquad \tau \ll \pi^2/(1 + 2/\gamma)^2, \quad \gamma \gg 1. \tag{23}$$

In the Tonks-Girardeau limit $\gamma \to \infty$, the above results provide $g^{(2)}(0) \to 0$ which indicates that two bosons cannot be found in the same spatial position (*antibunching* effect) as they are "fermionised" [38–40, 59, 76].

# 4 Equilibrium thermodynamics of a uniform 1D Bose gas

The complete finite-temperature thermodynamics in the canonical ensemble of a uniform 1D Bose gas, described by the Hamiltonian (1) is determined by the Helmholtz free energy $F$, which is obtained by using standard methods of statistical mechanics. All other thermodynamic quantities of interest can then be related to $F$ via an appropriate derivative [77].

An alternative approach, which is applicable in regimes III-VI of Fig. 1, relies on inverting the Hellmann-Feynman theorem wherein the Helmholtz free energy is calculated from known analytical expressions for the normalised local pair correlation function, $g^{(2)}(0)$, Eq. (13). We discuss this approach here. The local correlation function between two atoms $g^{(2)}(0)$ was evaluated for the first time in 2003 in Ref. [30], by using the exact thermal Bethe ansatz solution to the Yang-Yang thermodynamic equation for the free energy $F$ [27] and the subsequent numerical evaluation of its partial derivative with respect to the coupling constant $g$, according to,

$$g^{(2)}(0) = \frac{2}{Ln^2}\left(\frac{\partial F}{\partial g}\right)_{T,N,L}. \tag{24}$$

The trick presented in Eq. (24) is strictly applicable to a uniform gas and is essentially a finite-temperature extension of the Hellmann-Feynman theorem [78, 79] (see also Ref. [80] for a historical perspective), which expresses the derivative of the mean interaction energy of the Lieb-Liniger Hamiltonian $\langle \hat{H}_{\text{int}} \rangle$ in terms of the unnormalised pair correlation function, which is simply the numerator in Eq. (13):

$$\left(\frac{\partial F}{\partial g}\right)_{T,N,L} = \left(\frac{\partial \langle \hat{H}_{\text{int}} \rangle}{\partial g}\right)_{T,N,L} = \frac{\partial}{\partial g}\left[\frac{g}{2}\int_0^L dx \, \langle \hat{\psi}^\dagger(x)\hat{\psi}^\dagger(x)\hat{\psi}(x)\hat{\psi}(x)\rangle\right]_{T,N,L}$$
$$= \frac{L}{2}\langle \hat{\psi}^\dagger(x)\hat{\psi}^\dagger(x)\hat{\psi}(x)\hat{\psi}(x)\rangle. \tag{25}$$

While the numerical evaluation of $g^{(2)}(0)$ through the above trick, i.e., by knowing the exact TBA results for $F$, works for any interaction strength $\gamma$ and temperature $\tau$, approximate analytical expressions for $g^{(2)}(0)$ (holding only under certain limiting conditions on $\gamma$ and $\tau$) can be alternatively derived without resorting to any prior knowledge of $F$. Indeed, the local pair correlation $g^{(2)}(0)$, as well as the non-local pair correlation function,

$$g^{(2)}(r) = \frac{\langle \hat{\psi}^\dagger(x)\hat{\psi}^\dagger(x')\hat{\psi}(x')\hat{\psi}(x)\rangle}{n(x)n(x')}, \tag{26}$$

(where $r = |x - x'|$ is the relative distance between the spatial coordinates $x$ and $x'$) can also be evaluated directly from the expectation values of the bosonic field creation $\hat{\psi}^\dagger(x)$ and annihilation $\hat{\psi}(x)$ operators [30,35,36,40,59] using a variety of theoretical approaches. For example, the Bogoliubov theory which is applicable to a quasicondensate in the weakly-interacting regime (I and II regions in Fig. 1) [30,34,35,40,59,63,64], the Luttinger liquid description where the low-energy, long-wavelength phononic excitations rule the low-temperature thermodynamics along the whole interaction crossover (I and VI) [30,36,40,59,62,63], or perturbation theory using the path integral formalism with respect to $\gamma \ll 1$ (III and IV) or $1/\gamma \ll 1$ (V and VI) close to the limits of the ideal Bose gas ($\gamma = 0$) or the Tonks-Girardeau gas of infinitely strong interactions ($\gamma \to +\infty$), respectively [30,40]. In the latter case, perturbation theory with respect to $1/\gamma \ll 1$ (i.e. with large but finite interaction strength $\gamma \gg 1$) is equivalent to the perturbative approach for weakly-interacting spinless fermions, onto which the strongly-repulsive Bose gas can be mapped [73,74], just as for the case of the Bose-Fermi mapping in the strictly Tonks-Girardeau limit with $\gamma \to +\infty$ [6,35,63,64,81–84]. Thus, in Ref. [30], the local pair correlation function $g^{(2)}(0)$ was evaluated in two ways: (a) numerically and exactly,

using the Yang-Yang thermodynamics for the free energy $F$ and the Hellmann-Feynman trick (24), for any value of the interaction strength and temperature; and (b) analytically, by employing the corresponding approximations for the expectation values $\langle \hat{\psi}^\dagger(x)\hat{\psi}^\dagger(x)\hat{\psi}(x)\hat{\psi}(x)\rangle$ in Eq. (25) for the six different regimes of the 1D Bose gas, without relying on $F$.

Should approximate analytic approximations to $F$ have been known in 2003 in these limiting six regimes, the respective analytic limits to $g^{(2)}(0)$ could have been, of course, calculated via the above Hellmann-Feynman theorem as well. We note here, though, that such analytic expressions for $F$ have not been known in 2003 and have only been derived relatively recently by De Rosi *et al.* [62–64]. Even then, these recent results for $F$ cover only five (I, III, IV, V, and VI) out of the six regimes identified in the diagram of Fig. 1.

In the present work, we fill this gap and obtain approximate analytic limits for the Helmholtz free energy $F$ (as well as for other ensuing thermodynamic quantities) in regime II corresponding to the thermal quasicondensate. In addition, in regimes III and IV, relative to the degenerate and non-degenerate nearly ideal Bose gas, we find important additional terms in the thermodynamic properties, which couple the interaction strength with temperature. This is different from the existing literature, where the dependence on the interaction only appears at zero-temperature level [63, 64]. In the next Section 5, we show the validity of these analytic results as they exhibit an excellent agreement, within their range of applicability, with the exact (numerical) solutions following from the Yang-Yang thermodynamic Bethe ansatz method.

In regime II, we use the approximate analytic evaluation of the integrals in momentum space within the Bogoliubov theory for quasicondensates dominated by thermal (rather than quantum) fluctuations. Differentiating the resulting free energy $F$ with respect to the coupling constant $g$, as per the Hellmann-Feynman trick (24), reproduces the known result for the local pair correlation function $g^{(2)}(0)$ [30] and hence serves as a validity check of our novel finding for $F$. Moreover, by evaluating the same integrals under the opposite approximation of the Bogoliubov description where quantum (rather than thermal) fluctuations are more important, we recover the analytic finding for $F$ in regime I, obtained by De Rosi *et al.* [63, 64].

In regimes III and IV, we instead invert the Hellmann-Feynman trick (24) and employ the corresponding known analytic result for $g^{(2)}(0)$ [30] to evaluate $F(T, N, L, g)$. More specifically, this is achieved by integrating the $g^{(2)}(0)$ function with respect to the coupling constant $g$ from its value deep in the respective regime to $g = 0$ corresponding to the ideal (non-interacting) Bose gas limit, where the "integration constant" $F(T, N, L, g = 0)$ can be calculated directly from the known partition function for the ideal 1D Bose gas. It is worth noticing that the inverted trick (24) can be also applied to other regimes with the requirement that the integration range lies within a single region of Fig. 1, as the functional form of $g^{(2)}(0)$, in terms of the interaction strength and temperature, changes when crossing the regime boundaries.

The inverted Hellmann-Feynman trick is similar to the one used recently in Ref. [64] for regime IV, except that it was based on the known analytic expression for Tan's contact $\mathcal{C}$. This is not surprising, given that Tan's contact and the local pair correlation function $g^{(2)}(0)$ are directly proportional to each other [53, 54, 63, 64, 85–87],

$$\mathcal{C} = \frac{4m}{\hbar^2}\left(\frac{\partial F}{\partial a_{1D}}\right)_{T,N,L} = \frac{4nN}{a_{1D}^2}g^{(2)}(0),\tag{27}$$

with both being thermodynamic quantities. We notice, however, that our use of the inverted Hellmann-Feynman trick (24) does not rely on prior knowledge of Tan's contact. Instead, it only needs the knowledge of $g^{(2)}(0)$ itself, and as such, it can be easily extended to other analytically tractable regimes, such as V, and VI of Fig. 1. Indeed, by deriving the approximate limits for the free energy $F$ in these remaining two regimes in this way, we reproduce—as a validity check—the results of Ref. [64] (up to leading order expansion terms with respect to

relevant small parameters) calculated there using the virial (V) and Sommerfeld (VI) expansions of the ideal Fermi gas description where the interaction effects have been included with the hard-core "negative excluded volume" approach [64].

We further note that the inverted Hellmann-Feynman trick (24) cannot be applied in regime II because the "integration constant" in this case is unknown (as it corresponds to the free energy $F$ at the boundary between regions II and III, which is analytically inaccessible). Accordingly, our derivation of $F$ from the Bogoliubov theory in regime II is the only available option and highlights one of the key original results of this work. Similarly, the inverted Hellmann-Feynman trick is not applicable in regime I.

## 4.1 Quasicondensate regime

The thermodynamics of a one-dimensional weakly-repulsive Bose system at low temperatures is described using the extension of Bogoliubov theory to quasicondensates [34,35] in terms of a gas of non-interacting quasi-particle excitations satisfying bosonic statistics [63–65].

In the Bogoliubov treatment, the Lieb-Liniger Hamiltonian, Eq. (1), can be diagonalised to give

$$\hat{H}_{\mathrm{LL}} = E_0 + \sum_k \varepsilon(k)\hat{b}_k^\dagger \hat{b}_k\,, \tag{28}$$

where

$$\varepsilon(k) = \sqrt{\frac{\hbar^2 k^2}{2m}\left(\frac{\hbar^2 k^2}{2m} + 2gn\right)}\,, \tag{29}$$

is the Bogoliubov quasi-particle dispersion relation, $\hat{b}_k^\dagger$ and $\hat{b}_k$ are the creation and annihilation operators for the excitation modes, and

$$E_0 = \frac{g}{2}\frac{N^2}{L} + \frac{1}{2}\sum_k\left(\varepsilon(k) - \frac{\hbar^2 k^2}{2m} - gn\right) = N\frac{\hbar^2 n^2}{2m}\left(\gamma - \frac{4}{3\pi}\gamma^{3/2}\right)\,, \tag{30}$$

is the zero-temperature Lieb-Liniger ground-state energy in the Bogoliubov description [1]. Note that the allowed discretized momenta in the above sum (30) are given by $k = 2\pi j/L$ for all integers $j = 0, \pm 1, \pm 2, \ldots$ and the final result has been obtained by calculating the integral in momentum space in the thermodynamic limit ($N, L \to \infty$ at fixed 1D density $n = N/L$) and by employing Eq. (3).

Within the Bogoliubov theory, the Hamiltonian of the quasicondensate regime has been reduced to that of a system of independent (non-interacting) harmonic oscillators, Eq. (28) [65]. It follows that the calculation of the Helmholtz free energy $F$ and all other thermodynamic properties of the 1D Bose gas in this regime can proceed in the same way as for black-body radiation, using the Planck distribution for thermal occupancies of the Bogoliubov modes above the ground-state energy. The only difference is that the dispersion relation for photons, $\varepsilon(k) = \hbar k c = \hbar \omega_k$, must be replaced by the Bogoliubov spectrum, Eq. (29). As we will see below, this replacement leads to non-trivial integrals, the evaluation of which can be performed analytically in the two quasicondensate regimes I and II of Fig. 1.

In analogy with the thermodynamics of black-body radiation, the canonical partition function $Z$ for the entire many-body system is given by the product of all partition functions $Z_k$ for the harmonic oscillator modes of frequency $\omega_k = \varepsilon(k)/\hbar$, where each energy level $\varepsilon(k)$ is occupied by a number $s_k = 0, 1, 2, \ldots$ of excitation quanta, i.e., $Z = \prod_k Z_k$, where

$$Z_k = \sum_{s_k=0}^{\infty} e^{-s_k \varepsilon(k)/(k_B T)} = \frac{1}{1 - e^{-\varepsilon(k)/(k_B T)}}\,. \tag{31}$$

The Helmholtz free energy of the gas can then be obtained via $F = -k_B T \ln Z$, which is equivalent to summing the free energies $F_k = -k_B T \ln Z_k$ of each harmonic oscillator mode, $F = \sum_k F_k$. We note that this is the free energy associated only with the quasi-particle excitations, as it does not include the zero-temperature ground-state energy, $E_0$, given by Eq. (30). Adding $E_0$ to the thermal contribution of $F$ provided by excitations and going to the thermodynamic limit, we then find the total free energy

$$F = E_0 + \frac{k_B T L}{2\pi} \int_{-\infty}^{\infty} dk \, \ln\left(1 - e^{-\varepsilon(k)/(k_B T)}\right). \tag{32}$$

By integrating by parts, Eq. (32) can be simplified and the thermal free energy $F - E_0$ is expressed in terms of our dimensionless parameters for the interaction strength $\gamma$ (3) and temperature $\tau$ (4) as

$$F - E_0 = -N \frac{\hbar^2 n^2}{2m} \frac{\tau \sqrt{2\gamma}}{\pi} \int_0^{\infty} du \, \frac{\left(\sqrt{\tau^2 u^2/(4\gamma^2) + 1} - 1\right)^{1/2}}{e^u - 1}, \tag{33}$$

where $u = \epsilon(k)/(k_B T)$. It then suffices to evaluate the integral on the right-hand side of Eq. (33). This can be done either numerically in the entire quasicondensate regime (regions I and II) [63, 64] or analytically, which—as we will show below—can be performed separately under the respective conditions defining the two quasicondensate regimes I and II of Fig. 1.

### 4.1.1 Quantum quasicondensate regime (I)

We first analytically calculate the Helmholtz free energy $F$ in the quantum quasicondensate regime (region I), by introducing the dimensionless variable $x \equiv \tau u/(2\gamma)$ in the integral in the right-hand side of Eq. (33). We note that for $\tau \ll 2\gamma$, or equivalently $k_B T \ll gn$ (where $\tau = 2\gamma$ gives the approximate boundary between the regions I and II in Fig. 1), the main contribution to the integral comes from $x \ll 1$. The condition $x \ll 1$ can be well understood by considering that at low temperatures $k_B T \ll gn$ only quasi-particles with small momenta $\hbar|k| \ll m\sqrt{gn/m}$ are thermally excited (with $\hbar|k| = \sqrt{mk_B T}$ being the characteristic thermal momentum) and they correspond to phonons [62] entering in the leading contribution of the Bogoliubov spectrum $\varepsilon(k) \approx \hbar|k|\sqrt{gn/m}$, see Eq. (29). Since $u = \varepsilon(k)/(k_B T)$, $x \ll 1$ naturally emerges from the condition of small momenta. We can then make use, in Eq. (33), of the following Taylor series,

$$\left(\sqrt{x^2+1} - 1\right)^{1/2} = \frac{x}{\sqrt{2}} + \sum_{j=1}^{\infty} \frac{(4j-1)!}{(2j+1)!(2j-1)!} \frac{(-1)^j}{2^{4j-1/2}} x^{2j+1}, \tag{34}$$

valid for $0 \le x \le 1$, and which we prove in Appendix A. As the series diverges for $|x| > 1$, care must be taken when evaluating the integral in Eq. (33), which goes from $x = 0$ to $x = \infty$. In particular, we employ the above series in the integrand in Eq. (33) for all values $x > 0$, including $x > 1$, but then truncate the series at some finite order. This is permissible as for $x > 1$ sufficiently low-order terms in the series are suppressed by the large exponential term in the denominator of the integrand. The order to which the series is truncated can be raised for greater accuracy of the overall integral as we go deeper into the quantum quasicondensate regime $\tau \ll 2\gamma$ which implies $x \ll 1$, by ensuring the convergence of Eq. (34). Here, for definiteness and the sake of analytic transparency, we truncate the series at $j = 2$ (by keeping then three terms on the right-hand side of Eq. (34)), noting that additional corrections to the exact integral are negligible for $\tau \ll 2\gamma$. In this case, the integral in Eq. (33) can be evaluated easily, and the resulting Helmholtz free energy becomes

$$F = N \frac{\hbar^2 n^2}{2m} \left[ \gamma - \frac{4}{3\pi} \gamma^{3/2} - \frac{\pi}{12} \tau^2 \gamma^{-1/2} + \frac{\pi^3}{960} \tau^4 \gamma^{-5/2} - \frac{\pi^5}{4608} \tau^6 \gamma^{-9/2} \right], \tag{35}$$

where the zero-temperature limit $\tau = 0$ is provided by the ground-state energy (30).

From the free energy, we can calculate all other thermodynamic properties such as the pressure ($P$), the entropy ($S$), the chemical potential ($\mu$), the internal energy ($E$), the heat capacity at constant length ($C_L$), the inverse isothermal compressibility ($\kappa_T^{-1}$), and the local pair correlation function $\left[g^{(2)}(0)\right]$:

$$P = -\left(\frac{\partial F}{\partial L}\right)_{T,N,g} = \frac{\hbar^2 n^3}{2m}\left(\gamma - \frac{2}{3\pi}\gamma^{3/2} + \frac{\pi}{8}\tau^2\gamma^{-1/2} - \frac{7\pi^3}{1920}\tau^4\gamma^{-5/2} + \frac{11\pi^5}{9216}\tau^6\gamma^{-9/2}\right), \quad (36)$$

$$S = -\left(\frac{\partial F}{\partial T}\right)_{L,N,g} = Nk_B\left(\frac{\pi}{6}\tau\gamma^{-1/2} - \frac{\pi^3}{240}\tau^3\gamma^{-5/2} + \frac{\pi^5}{768}\tau^5\gamma^{-9/2}\right), \quad (37)$$

$$\mu = \left(\frac{\partial F}{\partial N}\right)_{T,L,g} = \frac{\hbar^2 n^2}{2m}\left(2\gamma - \frac{2}{\pi}\gamma^{3/2} + \frac{\pi}{24}\tau^2\gamma^{-1/2} - \frac{\pi^3}{384}\tau^4\gamma^{-5/2} + \frac{\pi^5}{1024}\tau^6\gamma^{-9/2}\right), \quad (38)$$

$$E = F + TS = N\frac{\hbar^2 n^2}{2m}\left(\gamma - \frac{4}{3\pi}\gamma^{3/2} + \frac{\pi}{12}\tau^2\gamma^{-1/2} - \frac{\pi^3}{320}\tau^4\gamma^{-5/2} + \frac{5\pi^5}{4608}\tau^6\gamma^{-9/2}\right), \quad (39)$$

$$C_L = \left(\frac{\partial E}{\partial T}\right)_{L,N,g} = Nk_B\left(\frac{\pi}{6}\tau\gamma^{-1/2} - \frac{\pi^3}{80}\tau^3\gamma^{-5/2} + \frac{5\pi^5}{768}\tau^5\gamma^{-9/2}\right), \quad (40)$$

$$\kappa_T^{-1} = n\left(\frac{\partial P}{\partial n}\right)_{T,N,g} = \frac{\hbar^2 n^3}{2m}\left(2\gamma - \frac{1}{\pi}\gamma^{3/2} - \frac{\pi}{16}\tau^2\gamma^{-1/2} + \frac{7\pi^3}{768}\tau^4\gamma^{-5/2} - \frac{11\pi^5}{2048}\tau^6\gamma^{-9/2}\right), \quad (41)$$

$$g^{(2)}(0) = \frac{2}{n^2 L}\left(\frac{\partial F}{\partial g}\right)_{T,N,L} = 1 - \frac{2}{\pi}\gamma^{1/2} + \frac{\pi}{24}\tau^2\gamma^{-3/2} - \frac{\pi^3}{384}\tau^4\gamma^{-7/2} + \frac{\pi^5}{1024}\tau^6\gamma^{-11/2}. \quad (42)$$

We note that these results are identical to those derived by De Rosi *et al.* [62–64] at the same level of approximation. We also recover results for the local pair correlation function $g^{(2)}(0)$, up to order $\mathcal{O}\left(\tau^2\right)$ [30], see Eq. (15), and $\mathcal{O}\left(\tau^4\right)$ [64], and we note that here we were able to improve upon the previous findings by obtaining higher-order expansion terms $\mathcal{O}\left(\tau^6\right)$ reported in Eq. (42).

### 4.1.2 Thermal quasicondensate regime (II)

In the thermal quasicondensate regime II of Fig. 1, we consider the approximation holding for $2\gamma \ll \tau \ll 2\sqrt{\gamma}$,

$$\int_0^\infty du \frac{\left(\sqrt{\tau^2 u^2/(4\gamma^2)+1}-1\right)^{1/2}}{e^u - 1} \approx \int_{2\gamma/\tau}^\infty du \frac{\sqrt{\tau u/(2\gamma)-1}}{e^u - 1} + \pi\left(1 - \frac{1}{\sqrt{2}}\right) - \frac{2\sqrt{2}\gamma}{3\tau}, \quad (43)$$

which we justify in Appendix B and for which the integration becomes restricted to $[2\gamma/\tau, \infty)$.

One then finds that the thermal contribution to the Helmholtz free energy (33) is captured by the high-momenta Bogoliubov quasiparticles:

$$F = N\frac{\hbar^2 n^2}{2m}\left(\gamma - \frac{\tau^{3/2}}{2\sqrt{\pi}}\mathrm{Li}_{3/2}\left(e^{-2\gamma/\tau}\right) - \left(\sqrt{2}-1\right)\tau\gamma^{1/2}\right)$$

$$\approx N\frac{\hbar^2 n^2}{2m}\left(\gamma - \frac{\zeta(3/2)}{2\sqrt{\pi}}\tau^{3/2} + \tau\gamma^{1/2} + \frac{\zeta(1/2)}{\sqrt{\pi}}\tau^{1/2}\gamma - \frac{\zeta(-1/2)}{\sqrt{\pi}}\tau^{-1/2}\gamma^2\right), \quad (44)$$

where $\zeta(z)$ is the Riemann zeta function and we have introduced the polylogarithm function $\mathrm{Li}_s(z) = \sum_{k=1}^\infty z^k/k^s$, valid for any complex order $s$ and complex argument $|z| < 1$. The polylogarithm function also admits the integral representation (often referred to as Bose function

or Bose-Einstein function),

$$\text{Li}_s(z) = \frac{1}{\Gamma(s)} \int_0^\infty dt \frac{t^{s-1}}{e^t/z - 1}, \tag{45}$$

with $\Gamma(s)$ the Euler gamma function. In Eq. (44) we have employed the following series holding for $z = e^{-\alpha} \approx 1$, with small and positive $\alpha = 2\gamma/\tau \ll 1$ [88]:[3]

$$\text{Li}_s\left(e^{-\alpha}\right) = \Gamma(1-s)\alpha^{s-1} + \sum_{n=0}^\infty (-1)^n \frac{\zeta(s-n)}{n!}\alpha^n, \tag{46}$$

and we have truncated the series at $n = 2$. To the best of our knowledge, the finite-temperature terms in the free energy derived in Eq. (44) were previously unknown. This is one of the main original results of the present paper.

From the free energy (44), we can then obtain—as we did previously for regime I—the pressure, the entropy, the chemical potential, the total internal energy, the heat capacity at constant length, the inverse isothermal compressibility, and the local pair correlation function:

$$P = \frac{\hbar^2 n^3}{2m}\left(\gamma + \frac{\zeta(3/2)}{2\sqrt{\pi}}\tau^{3/2} - \frac{1}{2}\tau\gamma^{1/2} - \frac{\zeta(-1/2)}{\sqrt{\pi}}\tau^{-1/2}\gamma^2\right), \tag{47}$$

$$S = Nk_B\left(\frac{3\zeta(3/2)}{4\sqrt{\pi}}\tau^{1/2} - \gamma^{1/2} - \frac{\zeta(1/2)}{2\sqrt{\pi}}\tau^{-1/2}\gamma - \frac{\zeta(-1/2)}{2\sqrt{\pi}}\tau^{-3/2}\gamma^2\right), \tag{48}$$

$$\mu = \frac{\hbar^2 n^2}{2m}\left(2\gamma + \frac{1}{2}\tau\gamma^{1/2} + \frac{\zeta(1/2)}{\sqrt{\pi}}\tau^{1/2}\gamma - \frac{2\zeta(-1/2)}{\sqrt{\pi}}\tau^{-1/2}\gamma^2\right), \tag{49}$$

$$E = N\frac{\hbar^2 n^2}{2m}\left(\gamma + \frac{\zeta(3/2)}{4\sqrt{\pi}}\tau^{3/2} + \frac{\zeta(1/2)}{2\sqrt{\pi}}\tau^{1/2}\gamma - \frac{3\zeta(-1/2)}{2\sqrt{\pi}}\tau^{-1/2}\gamma^2\right), \tag{50}$$

$$C_L = Nk_B\left(\frac{3\zeta(3/2)}{8\sqrt{\pi}}\tau^{1/2} + \frac{\zeta(1/2)}{4\sqrt{\pi}}\tau^{-1/2}\gamma + \frac{3\zeta(-1/2)}{4\sqrt{\pi}}\tau^{-3/2}\gamma^2\right), \tag{51}$$

$$\kappa_T^{-1} = \frac{\hbar^2 n^3}{2m}\left(2\gamma - \frac{1}{4}\tau\gamma^{1/2} - \frac{2\zeta(-1/2)}{\sqrt{\pi}}\tau^{-1/2}\gamma^2\right), \tag{52}$$

$$g^{(2)}(0) = 1 + \frac{1}{2}\tau\gamma^{-1/2} + \frac{\zeta(1/2)}{\sqrt{\pi}}\tau^{1/2} - \frac{2\zeta(-1/2)}{\sqrt{\pi}}\tau^{-1/2}\gamma. \tag{53}$$

The first two terms in the above equation for $g^{(2)}(0)$ reproduce the result of Ref. [30] (compare with Eq. (16)), whereas the last two terms are the new, higher-order temperature contributions derived here. All other analytical limits for the different thermodynamic quantities in regime II, Eqs. (44) and (47)–(52), are also novel findings of the present work.

The finite-temperature ($\tau \neq 0$) contributions in the thermodynamic properties prove to be not necessarily negligible here, as was the silent assumption prior to this work. As an example, the commonly used equation of state for the entire quasicondensate regime (i.e., in both regions, I and II) in the mean-field approximation is $P = \hbar^2 n^3 \gamma/(2m) = gn^2/2$ [compare Eqs. (36) and (47)], which is independent on temperature. Similarly, the mean-field chemical potential is $\mu = \hbar^2 n^2 \gamma/m = gn$ which is valid both in region I and II at $\mathcal{O}(\tau^0)$ level, see Eqs. (38) and (49). In contrast, our results for the same thermodynamic quantities are different for region I and II and provide then a fundamental tool to distinguish the physics emerging in the quantum and thermal quasicondensates. In addition, while in regime I ($\tau \ll 2\gamma$) the

---

[3]We note that the series in question given in Ref. [88] (see 1.11.8) is for the Lerch function $\Phi(z, s, v) = \sum_{k=0}^\infty z^k/(k+v)^s$. However, given that the polylogarithm function $\text{Li}_s(z)$ is a special case of the Lerch function for $v = 1$, i.e. $\text{Li}_s(z) = z\Phi(z, s, 1)$, the series (1.11.8) can be adopted for the polylogarithm function, resulting in Eq. (46).

finite-$\tau$ terms appear to be indeed negligible such that the thermodynamics is already well approximated at just the mean-field level, this is not necessarily the case in regime II ($\tau \gg 2\gamma$). In Sec. 5, we will show explicitly how the analytical limits for the thermodynamic properties including these finite-$\tau$ contributions agree with the exact thermal Bethe ansatz solution much better than the pure mean-field approximation, especially in regime II.

## 4.2 Nearly ideal Bose gas regime

To address the thermodynamics of the 1D nearly ideal ($\gamma \ll 1$) Bose gas system (regimes III and IV of Fig. 1), we first consider the simple limit of a non-interacting ($\gamma = 0$) or ideal Bose gas (IBG) at any temperature $\tau$, Eq. (4). Such a system can be treated using the standard formalism of the grand-canonical ensemble of statistical mechanics [77] and the recognition that the average values of all the thermodynamic quantities that we are interested in here will be the same as in the canonical ensemble, if calculated in the thermodynamic limit. The grand-canonical partition function for the 1D IBG is given by:

$$\mathcal{Z}_{\mathrm{IBG}} = \prod_k \frac{1}{1 - z e^{-\epsilon(k)/(k_B T)}} \,, \tag{54}$$

whereas the corresponding grand potential (or Landau potential), $\Omega = -k_B T \ln \mathcal{Z}$, by

$$\Omega_{\mathrm{IBG}} = k_B T \sum_k \ln\left(1 - z e^{-\epsilon(k)/(k_B T)}\right) . \tag{55}$$

Here, $z = e^{\mu/(k_B T)}$ is the fugacity and $\epsilon(k) = \hbar^2 k^2/(2m)$ denotes the free-particle dispersion relation with $k = (2\pi/L)j$ and integers $j = 0, \pm 1, \pm 2, ...$, for an uniform system with periodic boundary conditions and length $L$. Going into the thermodynamic limit, i.e., converting the discrete sum in Eq. (55) into a continuous integral $\sum_k \to L/(2\pi) \int_{-\infty}^{+\infty} dk$ and after some algebra and integration by parts, we recognise that $\Omega_{\mathrm{IBG}}$ can be defined in terms of the polylogarithm function $\mathrm{Li}_s(z)$ of Eq. (45), namely:

$$\Omega_{\mathrm{IBG}} = -\frac{L k_B T}{\lambda_T} \mathrm{Li}_{3/2}\left(e^{\mu/(k_B T)}\right) , \tag{56}$$

where $\lambda_T$ is the thermal de Broglie wavelength defined in Sec. 2.

The equation of state for the pressure $P$ of the IBG follows immediately from the thermodynamic relation $\Omega = -PL$ for uniform systems, giving:

$$P_{\mathrm{IBG}} = \frac{k_B T}{\lambda_T} \mathrm{Li}_{3/2}\left(e^{\mu/(k_B T)}\right) . \tag{57}$$

The partition function $\mathcal{Z}_{\mathrm{IBG}}$ or the grand potential $\Omega_{\mathrm{IBG}}$, which we note are functions of three parameters, $\mathcal{Z}_{\mathrm{IBG}} = \mathcal{Z}_{\mathrm{IBG}}(T, \mu, L)$ and $\Omega_{\mathrm{IBG}} = \Omega_{\mathrm{IBG}}(T, \mu, L)$, can be used to calculate the thermal average value of the total number of particles $\langle N \rangle_{\mathrm{IBG}} = (k_B T/\mathcal{Z}_{\mathrm{IBG}})(\partial \mathcal{Z}_{\mathrm{IBG}}/\partial \mu) = -\partial \Omega_{\mathrm{IBG}}/\partial \mu$ [77]. Alternatively, $\langle N \rangle_{\mathrm{IBG}}$ can be calculated using the Bose-Einstein distribution function for the mean occupancy of single-particle states and integrating it over all states. The average total number of particles, obtained in either of these alternative ways, gives the average 1D density, $\langle n \rangle_{\mathrm{IBG}} = \langle N \rangle_{\mathrm{IBG}}/L$, which can be expressed in terms of the polylogarithm function, Eq. (45), as follows:

$$\langle n \rangle_{\mathrm{IBG}} = \frac{1}{\lambda_T} \mathrm{Li}_{1/2}\left(e^{\mu/(k_B T)}\right) . \tag{58}$$

In the thermodynamic limit, the average density, $\langle n \rangle_{\mathrm{IBG}}$, takes the role of the fixed 1D density $n = N/L$ in the canonical ensemble where $N$ and $L$ are fixed (in addition to $T$). Therefore, by

inverting the above equation, and replacing $\langle n \rangle_{\text{IBG}} \to n$, we can solve it for $\mu$, hence obtaining the chemical potential of an IBG, $\mu_{\text{IBG}}$, as a function of fixed $n$:

$$\mu_{\text{IBG}} = \frac{\hbar^2 n^2}{2m} \tau \ln\left[ \text{Li}_{1/2}^{-1}\left( \frac{2\sqrt{\pi}}{\sqrt{\tau}} \right) \right], \tag{59}$$

where we have used the relation $\lambda_T n = 2\sqrt{\pi/\tau}$.

We can now apply the general thermodynamic relation $F = \Omega + \mu N$ to find that the Helmholtz free energy of an IBG in 1D is given by:

$$F_{\text{IBG}} = N\frac{\hbar^2 n^2}{2m}\left\{ \tau \ln\left[ \text{Li}_{1/2}^{-1}\left( \frac{2\sqrt{\pi}}{\sqrt{\tau}} \right) \right] - \frac{\tau^{3/2}}{2\sqrt{\pi}}\text{Li}_{3/2}\left[ \text{Li}_{1/2}^{-1}\left( \frac{2\sqrt{\pi}}{\sqrt{\tau}} \right) \right] \right\}, \tag{60}$$

whereas the equation of state for the pressure, Eq. (57), can now be rewritten as:

$$P_{\text{IBG}} = \frac{\hbar^2 n^3}{2m} \frac{\tau^{3/2}}{2\sqrt{\pi}}\text{Li}_{3/2}\left[ \text{Li}_{1/2}^{-1}\left( \frac{2\sqrt{\pi}}{\sqrt{\tau}} \right) \right]. \tag{61}$$

With the above expression for $F_{\text{IBG}}$ at hand, we can now calculate all other thermodynamic quantities of interest, as in Eqs. (37)–(42), by simply taking various partial derivatives of $F_{\text{IBG}}$.

### 4.2.1 Degenerate regime (III)

In the (quantum) degenerate regime of the nearly ideal Bose gas (region III of Fig. 1, where $2\sqrt{\gamma} \ll \tau \ll 1$), first-order perturbation theory about the ideal Bose gas $\gamma = 0$ yields the local pair correlation function $g^{(2)}(0)$, Eq. (19) [30].

We calculate here all the thermodynamic properties following from the $g^{(2)}(0)$. To this aim, we employ the inverted Hellmann-Feynman trick (24) and we integrate the local pair correlation function $g^{(2)}(0)$ to find the Helmholtz free energy $F(\gamma, \tau)$ at small and finite interaction strength $\gamma > 0$ [89]:

$$F(\gamma, \tau) = F_{\text{IBG}}(\tau) + N\frac{\hbar^2 n^2}{2m}\int_0^\gamma d\gamma'\, g^{(2)}(0; \gamma', \tau) = F_{\text{IBG}}(\tau) + N\frac{\hbar^2 n^2}{2m}\left( 2\gamma - \frac{2\gamma^2}{\tau^2} \right). \tag{62}$$

Here, we recognise the Helmholtz free energy of the ideal Bose gas, $F_{\text{IBG}}(\tau) \equiv F(\gamma = 0, \tau)$, as playing the role of the known integration constant [see Eq. (70) below]. Furthermore, in obtaining this result, we require that the functional form of the pair correlation $g^{(2)}(0) \equiv g^{(2)}(0; \gamma, \tau)$, as a function of $\gamma$ and $\tau$, Eq. (24), remains unchanged between the integration bounds in Eq. (62), hence the above result can be only applied to region III.

From the above expression (62), we find the pressure, the entropy, the chemical potential, the internal energy, the heat capacity at constant length, and the inverse isothermal compressibility:

$$P = -\left( \frac{\partial F}{\partial L} \right)_{T,N,g} = P_{\text{IBG}} + \frac{\hbar^2 n^3}{2m}\left( 2\gamma - \frac{8\gamma^2}{\tau^2} \right), \tag{63}$$

$$S = -\left( \frac{\partial F}{\partial T} \right)_{L,N,g} = S_{\text{IBG}} - Nk_B\left( \frac{4\gamma^2}{\tau^3} \right), \tag{64}$$

$$\mu = \left( \frac{\partial F}{\partial N} \right)_{T,L,g} = \mu_{\text{IBG}} + \frac{\hbar^2 n^2}{2m}\left( 4\gamma - 10\frac{\gamma^2}{\tau^2} \right), \tag{65}$$

$$E = F + TS = E_{\text{IBG}} + N\frac{\hbar^2 n^2}{2m}\left( 2\gamma - \frac{6\gamma^2}{\tau^2} \right), \tag{66}$$

$$C_L = \left(\frac{\partial E}{\partial T}\right)_{L,N,g} = C_{L,\mathrm{IBG}} + N k_B \left(\frac{12\gamma^2}{\tau^3}\right), \tag{67}$$

$$\kappa_T^{-1} = n\left(\frac{\partial P}{\partial n}\right)_{T,N,g} = \kappa_{T,\mathrm{IBG}}^{-1} + \frac{\hbar^2 n^3}{2m}\left(4\gamma - \frac{40\gamma^2}{\tau^2}\right). \tag{68}$$

In the findings (62)-(68), the terms that are only dependent on the interaction strength $\gamma$, reproduce the zero-temperature results in the Hartree-Fock approximation [64]. The local pair correlation function in the Hartree-Fock theory coincides with that of the ideal Bose gas $g^{(2)}(0) = 2$, as it does not contain any finite-$\gamma$ corrections. On the other hand, the contribution $\mathcal{O}(\gamma/\tau^2)$ in the $g^{(2)}(0)$ employed here, Eq. (19), generates next-order corrections in the thermodynamic properties, (62)-(68), which have not been previously reported, to the best of our knowledge.

In the above analytical approximations for the thermodynamic quantities, the corresponding highly-degenerate ($\tau \ll 1$) limits for the ideal Bose gas can be derived from Eq. (60) using the series expansion of Eq. (46). We set $2\sqrt{\pi/\tau} \equiv x = \mathrm{Li}_s(e^{-\alpha})$, so that $\mathrm{Li}_s^{-1}(x) = e^{-\alpha}$ and restrict ourselves to the first two terms in the expansion (46), i.e. the first term and the $n=0$ term from the sum. One then obtains that $\alpha \simeq [(x - \zeta(s))/\Gamma(1-s)]^{1/(s-1)}$, and therefore $\mathrm{Li}_s^{-1}(x) \simeq \exp\{-[(x - \zeta(s))/\Gamma(1-s)]^{1/(s-1)}\}$. This in turn implies

$$\mathrm{Li}_s^{-1}\left(\frac{2\sqrt{\pi}}{\sqrt{\tau}}\right) \simeq \exp\left[-\left(\frac{2\sqrt{\pi/\tau} - \zeta(s)}{\Gamma(1-s)}\right)^{\frac{1}{s-1}}\right]. \tag{69}$$

Using Eq. (69), the Helmholtz free energy of a highly-degenerate IBG, from Eq. (60), can then be shown to be given by:

$$F_{\mathrm{IBG}} = N\frac{\hbar^2 n^2}{2m}\frac{\tau^{3/2}}{2\sqrt{\pi}}\left[-\zeta(3/2) + \frac{\pi}{\frac{2\sqrt{\pi}}{\sqrt{\tau}} - \zeta(1/2)}\right] + \mathcal{O}(\tau^{5/2}). \tag{70}$$

From this, we obtain the following results for all other thermodynamic quantites of interest:

$$P_{\mathrm{IBG}} = \frac{\hbar^2 n^3}{2m}\frac{\tau^{3/2}}{2\sqrt{\pi}}\left[\zeta(3/2) - \pi\frac{\frac{4\sqrt{\pi}}{\sqrt{\tau}} - \zeta(1/2)}{\left(\frac{2\sqrt{\pi}}{\sqrt{\tau}} - \zeta(1/2)\right)^2}\right], \tag{71}$$

$$S_{\mathrm{IBG}} = N k_B \frac{\sqrt{\tau}}{2\sqrt{\pi}}\left[\frac{3}{2}\zeta(3/2) - \frac{\pi}{2}\frac{\frac{8\sqrt{\pi}}{\sqrt{\tau}} - 3\zeta(1/2)}{\left(\frac{2\sqrt{\pi}}{\sqrt{\tau}} - \zeta(1/2)\right)^2}\right], \tag{72}$$

$$\mu_{\mathrm{IBG}} = -\frac{\hbar^2 n^2}{2m}\frac{\pi\tau}{\left(\frac{2\sqrt{\pi}}{\sqrt{\tau}} - \zeta(1/2)\right)^2}, \tag{73}$$

$$E_{\mathrm{IBG}} = N\frac{\hbar^2 n^2}{2m}\frac{\tau^{3/2}}{2\sqrt{\pi}}\left[\frac{1}{2}\zeta(3/2) - \frac{\pi}{2}\frac{\frac{4\sqrt{\pi}}{\sqrt{\tau}} - \zeta(1/2)}{\left(\frac{2\sqrt{\pi}}{\sqrt{\tau}} - \zeta(1/2)\right)^2}\right], \tag{74}$$

$$C_{L,\mathrm{IBG}} = N k_B \frac{\sqrt{\tau}}{2\sqrt{\pi}}\left[\frac{3}{4}\zeta(3/2) - \frac{\pi}{2}\frac{\frac{16\pi}{\tau} - \frac{9\sqrt{\pi}}{\sqrt{\tau}}\zeta(1/2) + \frac{3}{2}\zeta(1/2)^2}{\left(\frac{2\sqrt{\pi}}{\sqrt{\tau}} - \zeta(1/2)\right)^3}\right], \tag{75}$$

$$\kappa_{T,\text{IBG}}^{-1} = \frac{\hbar^2 n^3}{2m} \frac{\sqrt{\tau}}{2\sqrt{\pi}} \frac{8\pi^2}{\left(\frac{2\sqrt{\pi}}{\sqrt{\tau}} - \zeta(1/2)\right)^3}. \tag{76}$$

An analogous low-temperature expansion was performed within the Hartree-Fock theory in Ref. [64], where the IBG case is recovered for zero coupling constant $g = 0$.

### 4.2.2 Non-degenerate regime (IV)

In the (classical) non-degenerate regime $\tau \gg \max\{1, \gamma^2\}$ (region IV in Fig. 1), approaching the Maxwell-Boltzmann gas limit $\gamma = 0$ at high temperatures, the local pair correlation function $g^{(2)}(0)$ takes the form, Eq. (20) [30].

Similarly to what we did in Eq. (62), we integrate $g^{(2)}(0)$ to recover the Helmholtz free energy at finite interaction strength $\gamma > 0$ within the non-degenerate regime [89]:

$$F = F_{\text{IBG}} + N \frac{\hbar^2 n^2}{2m} \left(2\gamma - \gamma^2 \sqrt{\frac{\pi}{2\tau}}\right). \tag{77}$$

From this, we obtain the pressure, the entropy, the chemical potential, the total internal energy, the heat capacity at constant length, and the inverse isothermal compressibility:

$$P = P_{\text{IBG}} + \frac{\hbar^2 n^3}{2m} \left(2\gamma - \gamma^2 \sqrt{\frac{\pi}{2\tau}}\right), \tag{78}$$

$$S = S_{\text{IBG}} - N k_B \left(\frac{1}{2} \sqrt{\frac{\pi}{2\tau^3}} \gamma^2\right), \tag{79}$$

$$\mu = \mu_{\text{IBG}} + \frac{\hbar^2 n^2}{2m} \left(4\gamma - \gamma^2 \sqrt{\frac{2\pi}{\tau}}\right), \tag{80}$$

$$E = E_{\text{IBG}} + N \frac{\hbar^2 n^2}{2m} \left(2\gamma - \frac{3}{2} \sqrt{\frac{\pi}{2\tau}} \gamma^2\right), \tag{81}$$

$$C_L = C_{L,\text{IBG}} + N k_B \left(\frac{3}{4} \sqrt{\frac{\pi}{2\tau^3}} \gamma^2\right), \tag{82}$$

$$\kappa_T^{-1} = \kappa_{T,\text{IBG}}^{-1} + \frac{\hbar^2 n^3}{2m} \left(4\gamma - \gamma^2 \sqrt{\frac{2\pi}{\tau}}\right). \tag{83}$$

Similarly to the results for the degenerate regime III, the first-order terms $\mathcal{O}(\gamma)$ in the interaction strength $\gamma$ in the above expressions recover the zero-temperature limit of the Hartree-Fock theory [63, 64]. The second-order contributions $\mathcal{O}(\gamma^2)$, on the other hand, have not been previously reported, to the best of our knowledge.

In the above analytical limits for the various thermodynamic properties, the nondegenerate ($\tau \gg 1$) ideal Bose gas results can be derived by expanding the Helmholtz free energy, Eq. (60), in powers of the small parameter $x = 2\sqrt{\pi/\tau} \ll 1$ using the series expansion of the inverse of the polylogarithm function, $\text{Li}_{1/2}^{-1}(x)$, to the desired order. Such an expansion, in general, can be shown to be given by $\text{Li}_s^{-1}(x) = \sum_{k=1}^{\infty} a_k x^k$, where $a_1 = 1$, $a_2 = -2^{-s}$, $a_3 = 2^{1-2s} - 3^{-s}$, $a_4 = 56^{-s} - 8^{-s}(5 + 2^s)$, $\cdots$ [90]. The Helmholtz free energy for the nondegenerate IBG is then given by, up to the fourth-order terms ($\propto 1/\sqrt{\tau}$):

$$F_{\text{IBG}} = N \frac{\hbar^2 n^2}{2m} \left[\tau \ln\left(\frac{2\sqrt{\pi}}{\sqrt{\tau}}\right) - \tau - \sqrt{\frac{\pi\tau}{2}} + \pi\left(1 - \frac{4}{3\sqrt{3}}\right) + \pi^{3/2} \frac{(2\sqrt{3} - 5)}{3\sqrt{2\tau}}\right]. \tag{84}$$

All other thermodynamic quantities that follow from this $F_{\text{IBG}}$ are given by:

$$P_{\text{IBG}} = \frac{\hbar^2 n^3}{2m}\left[\tau - \sqrt{\frac{\pi\tau}{2}} + 2\pi\left(1 - \frac{4}{3\sqrt{3}}\right) + \pi^{3/2}\frac{\left(2\sqrt{3}-5\right)}{\sqrt{2\tau}}\right], \tag{85}$$

$$S_{\text{IBG}} = N k_B\left[\ln\left(\frac{\sqrt{\tau}}{2\sqrt{\pi}}\right) + \frac{3}{2} + \frac{\sqrt{\pi}}{2\sqrt{2\tau}} + \pi^{3/2}\frac{2\sqrt{3}-5}{6\sqrt{2\tau^3}}\right], \tag{86}$$

$$\mu_{\text{IBG}} = \frac{\hbar^2 n^2}{2m}\left[\tau\ln\left(\frac{2\sqrt{\pi}}{\sqrt{\tau}}\right) - \sqrt{2\pi\tau} + 3\pi\left(1 - \frac{4}{3\sqrt{3}}\right) + 4\pi^{3/2}\frac{\left(2\sqrt{3}-5\right)}{3\sqrt{2\tau}}\right], \tag{87}$$

$$E_{\text{IBG}} = N\frac{\hbar^2 n^2}{2m}\left[\frac{\tau}{2} - \frac{1}{2}\sqrt{\frac{\pi\tau}{2}} + \pi\left(1 - \frac{4}{3\sqrt{3}}\right) + \pi^{3/2}\frac{2\sqrt{3}-5}{2\sqrt{2\tau}}\right], \tag{88}$$

$$C_{L,\text{IBG}} = N k_B\left[\frac{1}{2} - \frac{1}{4}\sqrt{\frac{\pi}{2\tau}} - \pi^{3/2}\frac{2\sqrt{3}-5}{4\sqrt{2\tau^3}}\right], \tag{89}$$

$$\kappa_{T,\text{IBG}}^{-1} = \frac{\hbar^2 n^3}{2m}\left[\tau - \sqrt{2\pi\tau} + 6\pi\left(1 - \frac{4}{3\sqrt{3}}\right) + 2\sqrt{2}\pi^{3/2}\frac{\left(2\sqrt{3}-5\right)}{\sqrt{\tau}}\right]. \tag{90}$$

In the above results for the ideal Bose gas case, we notice that the leading-order (first) terms in the temperature $\tau$, correspond to the familiar textbook results for the Maxwell-Boltzmann classical ideal gas. Moreover, the first two contributions in the entropy $S_{\text{IBG}}$, Eq. (86), reproduce the Sackur-Tetrode equation describing the classical gas [77].

A similar high-temperature virial expansion has been presented in Refs. [63, 64] in the framework of the Hartree-Fock theory, where the non-degenerate IBG limit is recovered for zero coupling constant $g = 0$.

## 4.3 Strongly-interacting regime

We discuss here the strongly-interacting regime (regions V and VI of Fig. 1). A special limit is provided by infinite interaction strength, $\gamma \to \infty$, where the 1D Bose system enters the so-called Tonks-Girardeau regime [6,68] and can be treated as a spin polarised 1D ideal Fermi gas (IFG) via the Bose-Fermi mapping [35,81–84], resulting in identical thermodynamics [63, 64]. We can then study the Tonks-Girardeau limit similarly as we did in Sec. 4.2 for the ideal Bose gas, with the only difference that the grand-canonical partition function $\mathcal{Z}$ and the grand potential $\Omega$ are now given by

$$\mathcal{Z}_{\text{IFG}} = \prod_k\left[1 + z e^{-\epsilon(k)/(k_B T)}\right], \tag{91}$$

and

$$\Omega_{\text{IFG}} = -k_B T \sum_k \ln\left(1 + z e^{-\epsilon(k)/(k_B T)}\right), \tag{92}$$

where $z = e^{\mu/(k_B T)}$ is the fugacity. Furthermore, when calculating the thermal average of the total number of particles $\langle N\rangle_{\text{IFG}}$ for evaluating the average 1D linear density $\langle n\rangle_{\text{IFG}} = \langle N\rangle_{\text{IFG}}/L$, we replace the Bose-Einstein distribution function with the Fermi-Dirac distribution function.

By going to the thermodynamic limit and carrying out the relevant integrals, we obtain the following results for the grand potential $\Omega_{\text{IFG}}$, the pressure $P_{\text{IFG}}$, and the average particle number density $\langle n\rangle_{\text{IFG}}$ of the IFG:

$$\Omega_{\text{IFG}} = \frac{L k_B T}{\lambda_T}\text{Li}_{3/2}\left(-e^{\mu/(k_B T)}\right), \tag{93}$$

$$P_{\text{IFG}} = -\frac{k_B T}{\lambda_T} \text{Li}_{3/2}\left(-e^{\mu/(k_B T)}\right), \tag{94}$$

$$\langle n \rangle_{\text{IFG}} = -\frac{1}{\lambda_T} \text{Li}_{1/2}\left(-e^{\mu/(k_B T)}\right). \tag{95}$$

Inverting the last expression, and replacing $\langle n \rangle_{\text{IFG}} \to n$, we find the chemical potential

$$\mu_{\text{IFG}} = \frac{\hbar^2 n^2}{2m} \tau \ln\left[-\text{Li}_{1/2}^{-1}\left(-\frac{2\sqrt{\pi}}{\sqrt{\tau}}\right)\right]. \tag{96}$$

The Helmholtz free energy $F = \Omega + \mu N$ for the ideal Fermi gas is then given by

$$F_{\text{IFG}} = N\frac{\hbar^2 n^2}{2m}\left\{\tau \ln\left[-\text{Li}_{1/2}^{-1}\left(-\frac{2\sqrt{\pi}}{\sqrt{\tau}}\right)\right] + \frac{\tau^{3/2}}{2\sqrt{\pi}}\text{Li}_{3/2}\left[\text{Li}_{1/2}^{-1}\left(-\frac{2\sqrt{\pi}}{\sqrt{\tau}}\right)\right]\right\}, \tag{97}$$

whereas the above equation of state for the pressure becomes gives:

$$P_{\text{IFG}} = -\frac{\hbar^2 n^3}{4m}\frac{\tau^{3/2}}{\sqrt{\pi}}\text{Li}_{3/2}\left[\text{Li}_{1/2}^{-1}\left(-\frac{2\sqrt{\pi}}{\sqrt{\tau}}\right)\right]. \tag{98}$$

Close to the Tonks-Girardeau limit $\gamma \to \infty$, at large but finite interaction strengths, $\gamma \gg 1$, the thermodynamics of the 1D Bose gas can be calculated at any temperature by relying on the hard-core (HC) model [63, 64, 91], where the size of the IFG system $L$ is modified by the "negative excluded volume" $Na_{1D}$ occupied by $N$ spheres whose diameter is provided by the 1D $s-$wave scattering length $a_{1D} < 0$:

$$L \to L - Na_{1D} = L(1 + 2/\gamma), \tag{99}$$

where we have used Eq. (3). The Helmholtz free energy in the hard-core model can then be obtained by a simple substitution of the modified system size, Eq. (99), into Eq. (97):

$$F(\gamma) = F_{\text{IFG}}\big[L \to L(1 + 2/\gamma)\big]. \tag{100}$$

From this, one can then calculate all the other remaining thermodynamic quantities as usual [as was done in Eqs. (36)–(42)], including the local pair correlation function $g^{(2)}(0)$. An alternative approach would be to develop the perturbation theory for small parameter $1/\gamma$ [73, 74], as it was implemented for the calculation of $g^{(2)}(0)$ in Ref. [30], however, we do not pursue this second option here.

### 4.3.1 High-temperature fermionization regime (V)

We discuss here the high-temperature (non-degenerate) regime of a strongly interacting 1D Bose gas, corresponding to region V and $\pi^2/(1 + 2/\gamma)^2 \ll \tau \ll \gamma^2$ in Fig. 1.

In the Tonks-Girardeau limit $\gamma \to \infty$, the thermodynamics of the non-degenerate regime $\tau \gg \pi^2$ is well described by the Helmholtz free energy of the non-degenerate 1D ideal Fermi gas. The latter can be obtained as in region IV for a nondegenerate ideal Bose gas, i.e., using a direct series expansion of the polylogarithm functions in Eq. (97) in terms of the small argument $2\sqrt{\pi/\tau} \ll 1$. The resulting expression for the high-temperature free energy is nearly identical to that in Eq. (84) for the IBG, except the opposite sign in front of the third term:

$$F_{\text{IFG}} = N\frac{\hbar^2 n^2}{2m}\left[\tau \ln\left(\frac{2\sqrt{\pi}}{\sqrt{\tau}}\right) - \tau + \sqrt{\frac{\pi\tau}{2}} + \pi\left(1 - \frac{4}{3\sqrt{3}}\right) + \pi^{3/2}\frac{2\sqrt{3} - 5}{3\sqrt{2\tau}}\right]. \tag{101}$$

We note that this result reproduces the one derived recently using a high-temperature virial expansion [63, 64] similarly to the non-degenerate IBG findings for regime IV.

To treat the strongly-interacting regime of very large but finite dimensionless interaction strength, $\gamma \gg 1$, we implement the hard-core model substitution (100) in the IFG free energy (101) and find:

$$F = F_{\text{IFG}} + N\frac{\hbar^2 n^2}{2m}\left[-\frac{2\tau}{\gamma} - \frac{\sqrt{2\pi\tau}}{\gamma} - \frac{4\pi}{\gamma}\left(1 - \frac{4}{3\sqrt{3}}\right) + \frac{2\tau}{\gamma^2} + \frac{2\sqrt{2\pi\tau}}{\gamma^2} - \frac{8\tau}{3\gamma^3}\right]. \quad (102)$$

Here, we have performed a further series expansion for $\gamma \to \infty$ and we have retained only the dominant terms for high temperatures $\tau \to \infty$.

One can then obtain the pressure, the entropy, the chemical potential, the total internal energy, the heat capacity at constant length, the inverse isothermal compressibility, and the local pair correlation function, as previously (as in regime I):

$$P = P_{\text{IFG}} + \frac{\hbar^2 n^3}{2m}\left[-\frac{2\tau}{\gamma} - \frac{2\sqrt{2\pi\tau}}{\gamma} - \frac{4\pi}{\gamma}\left(3 - \frac{4}{\sqrt{3}}\right) + \frac{4\tau}{\gamma^2} + \frac{6\sqrt{2\pi\tau}}{\gamma^2} - \frac{8\tau}{\gamma^3}\right], \quad (103)$$

$$S = S_{\text{IFG}} + Nk_B\left[\frac{2}{\gamma} + \frac{\sqrt{\pi}}{\gamma\sqrt{2\tau}} - \frac{2}{\gamma^2} - \frac{\sqrt{2\pi}}{\gamma^2\sqrt{\tau}} + \frac{8}{3\gamma^3}\right], \quad (104)$$

$$\mu = \mu_{\text{IFG}} + \frac{\hbar^2 n^2}{2m}\left[-\frac{4\tau}{\gamma} - \frac{3\sqrt{2\pi\tau}}{\gamma} - \frac{8\pi}{\gamma}\left(2 - \frac{8}{3\sqrt{3}}\right) + \frac{6\tau}{\gamma^2} + \frac{8\sqrt{2\pi\tau}}{\gamma^2} - \frac{32\tau}{3\gamma^3}\right], \quad (105)$$

$$E = E_{\text{IFG}} + N\frac{\hbar^2 n^2}{2m}\left[-\frac{\sqrt{2\pi\tau}}{2\gamma} - \frac{4\pi}{\gamma}\left(1 - \frac{4}{3\sqrt{3}}\right) + \frac{\sqrt{2\pi\tau}}{\gamma^2}\right], \quad (106)$$

$$C_L = C_{L,\text{IFG}} + Nk_B\left[-\frac{\sqrt{2\pi}}{4\gamma\sqrt{\tau}} + \frac{\sqrt{2\pi}}{2\gamma^2\sqrt{\tau}}\right], \quad (107)$$

$$\kappa_T^{-1} = \kappa_{T,\text{IFG}}^{-1} + \frac{\hbar^2 n^3}{2m}\left[-\frac{4\tau}{\gamma} - \frac{6\sqrt{2\pi\tau}}{\gamma} - \frac{16\pi}{\gamma}\left(3 - \frac{4}{\sqrt{3}}\right) + \frac{12\tau}{\gamma^2} + \frac{24\sqrt{2\pi\tau}}{\gamma^2} - \frac{32\tau}{\gamma^3}\right], \quad (108)$$

$$g^{(2)}(0) = \frac{2\tau}{\gamma^2} + \frac{\sqrt{2\pi\tau}}{\gamma^2} + \frac{4\pi}{\gamma^2}\left(1 - \frac{4}{3\sqrt{3}}\right) - \frac{4\tau}{\gamma^3} - \frac{4\sqrt{2\pi\tau}}{\gamma^3} + \frac{8\tau}{\gamma^4}. \quad (109)$$

The first term $\mathcal{O}\left(\tau/\gamma^2\right)$ in the local pair correlation function $g^{(2)}(0)$ above, Eq. (109), was originally derived via second-order perturbation theory for small $1/\gamma$ parameter approaching the Tonks-Girardeau limit $\gamma \to \infty$ [30], Eq. (22). Here, by instead including the interaction effects via the hard-core "negative excluded volume" corrections [63, 64], we were able to obtain higher-order contributions, by considering up to order $\mathcal{O}\left(1/\gamma^4\right)$ for a better agreement with the thermal Bethe ansatz exact results (see Sec. 5).

Some of the other thermodynamic quantities have been previously derived with the hard-core model without any series expansion for large $\gamma$ and $\tau$ [63, 64], in contrast to the present results like the free energy (102). However, differently from the literature [63, 64], the analytical approximations reported here reveal a clear dependence on the key parameters $\gamma$ and $\tau$ of each dominant term in the thermodynamic properties, while still exhibiting a fair agreement with the thermal Bethe-ansatz solution, as shown in Sec. 5.

In the above expressions, the respective results for the ideal Fermi gas in the non-degenerate regime ($\tau \gg \pi^2$) can be obtained via a high-temperature virial expansion [63, 64]:

$$P_{\text{IFG}} = \frac{\hbar^2 n^3}{2m}\left[\tau + \sqrt{\frac{\pi\tau}{2}} + 2\pi\left(1 - \frac{4}{3\sqrt{3}}\right) + \pi^{3/2}\frac{\left(2\sqrt{3} - 5\right)}{\sqrt{2\tau}}\right], \quad (110)$$

$$S_{\text{IFG}} = Nk_B\left[\ln\left(\frac{\sqrt{\tau}}{2\sqrt{\pi}}\right) + \frac{3}{2} - \frac{\sqrt{\pi}}{2\sqrt{2\tau}} + \pi^{3/2}\frac{2\sqrt{3} - 5}{6\sqrt{2\tau^3}}\right], \quad (111)$$

$$\mu_{\text{IFG}} = \frac{\hbar^2 n^2}{2m}\left[\tau \ln\left(\frac{2\sqrt{\pi}}{\sqrt{\tau}}\right) + \sqrt{2\pi\tau} + 3\pi\left(1 - \frac{4}{3\sqrt{3}}\right) + 4\pi^{3/2}\frac{(2\sqrt{3}-5)}{3\sqrt{2\tau}}\right], \tag{112}$$

$$E_{\text{IFG}} = N\frac{\hbar^2 n^2}{2m}\left[\frac{\tau}{2} + \frac{1}{2}\sqrt{\frac{\pi\tau}{2}} + \pi\left(1 - \frac{4}{3\sqrt{3}}\right) + \pi^{3/2}\frac{2\sqrt{3}-5}{2\sqrt{2\tau}}\right], \tag{113}$$

$$C_{L,\text{IFG}} = Nk_B\left[\frac{1}{2} + \frac{1}{4}\sqrt{\frac{\pi}{2\tau}} - \pi^{3/2}\frac{2\sqrt{3}-5}{4\sqrt{2\tau^3}}\right], \tag{114}$$

$$\kappa_{T,\text{IFG}}^{-1} = \frac{\hbar^2 n^3}{2m}\left[\tau + \sqrt{2\pi\tau} + 6\pi\left(1 - \frac{4}{3\sqrt{3}}\right) + 2\sqrt{2}\pi^{3/2}\frac{(2\sqrt{3}-5)}{\sqrt{\tau}}\right]. \tag{115}$$

By comparing the thermodynamic properties for the ideal Fermi gas at high temperatures, Eqs. (101), (110)-(115), with the corresponding results for the ideal Bose gas reported in SubSec. 4.2.2, we notice that they share the same leading-order terms describing the Maxwell-Boltzmann classical gas. The next-order perturbative contribution exhibits instead an opposite sign emerging from the quantum statistics, whose effects become more important at lower temperatures.

Despite the existence of the fermionized regime in the 1D Bose gas, which is absent in 2D and 3D systems, we note that, for any fixed small or large value of the dimensionless interaction strength $\gamma$, the system becomes described by the Maxwell-Boltzmann theory of the classical ideal gas by increasing the temperature $\tau$. This can be easily seen from the diagram of regimes in Fig. 1 (a) by tracing a vertical line at any fixed $\gamma$. At large enough values of $\tau$, the gas necessarily enters regime IV corresponding to the non-degenerate nearly ideal Bose gas. By further raising the temperature within this regime, the system can be eventually described by the classical ideal gas model. Indeed, in the limit of very high temperatures, the perturbative terms depending on $\gamma$ in the thermodynamic properties of regime IV in SubSec. 4.2.2 become negligible compared to the leading $\tau$-dependent contributions of the ideal Bose gas, hence recovering the expected classical ideal gas result.

### 4.3.2 Low-temperature fermionization regime (VI)

In the low-temperature, strongly-interacting regime (region VI in Fig. 1), defined via the conditions $\tau \ll \pi^2/(1 + 2/\gamma)^2$ and $\gamma \gg 1$, we again apply the hard-core "negative excluded volume" model to the known result for the 1D ideal Fermi gas.

The Helmholtz free energy in the Tonks-Girardeau limit $\gamma \to \infty$ at low temperatures $\tau \ll \pi^2$ can be obtained with a Sommerfeld expansion [63, 64, 92], yielding:

$$F_{\text{IFG}} = N\frac{\hbar^2 n^2}{2m}\left(\frac{\pi^2}{3} - \frac{\tau^2}{12} - \frac{\tau^4}{180\pi^2} - \frac{7\tau^6}{1296\pi^4}\right). \tag{116}$$

For large but finite interaction strength $\gamma \gg 1$, we consider the "negative excluded volume" prescription (100) in the ideal Fermi gas free energy, Eq. (116), and derive

$$F = F_{\text{IFG}} + N\frac{\hbar^2 n^2}{2m}\left(-\frac{4\pi^2}{3\gamma} - \frac{\tau^2}{3\gamma} + \frac{4\pi^2}{\gamma^2} - \frac{\tau^2}{3\gamma^2} - \frac{32\pi^2}{3\gamma^3}\right), \tag{117}$$

which has been obtained with an additional series expansion at large interaction strength $\gamma \to \infty$ and low temperatures $\tau \to 0$. From the free energy (117) and as we did in subsection 4.3.1, we calculate the pressure, the entropy, the chemical potential, the total internal energy, the heat capacity at constant length, the inverse isothermal compressibility, and the local pair correlation function:

$$P = P_{\text{IFG}} + \frac{\hbar^2 n^3}{2m}\left(-\frac{4\pi^2}{\gamma} + \frac{\tau^2}{3\gamma} + \frac{16\pi^2}{\gamma^2} - \frac{160\pi^2}{3\gamma^3}\right), \tag{118}$$

$$S = S_{\text{IFG}} + Nk_B \left( \frac{2\tau}{3\gamma} + \frac{2\tau}{3\gamma^2} \right), \tag{119}$$

$$\mu = \mu_{\text{IFG}} + \frac{\hbar^2 n^2}{2m} \left( -\frac{16\pi^2}{3\gamma} + \frac{20\pi^2}{\gamma^2} - \frac{\tau^2}{3\gamma^2} - \frac{64\pi^2}{\gamma^3} \right), \tag{120}$$

$$E = E_{\text{IFG}} + N\frac{\hbar^2 n^2}{2m} \left( -\frac{4\pi^2}{3\gamma} + \frac{\tau^2}{3\gamma} + \frac{4\pi^2}{\gamma^2} + \frac{\tau^2}{3\gamma^2} - \frac{32\pi^2}{3\gamma^3} \right), \tag{121}$$

$$C_L = C_{L,\text{IFG}} + Nk_B \left( \frac{2\tau}{3\gamma} + \frac{2\tau}{3\gamma^2} \right), \tag{122}$$

$$\kappa_T^{-1} = \kappa_{T,\text{IFG}}^{-1} + \frac{\hbar^2 n^3}{2m} \left( -\frac{16\pi^2}{\gamma} + \frac{80\pi^2}{\gamma^2} - \frac{320\pi^2}{\gamma^3} \right), \tag{123}$$

$$g^{(2)}(0) = \frac{4\pi^2}{3\gamma^2} + \frac{\tau^2}{3\gamma^2} - \frac{8\pi^2}{\gamma^3} + \frac{2\tau^2}{3\gamma^3} + \frac{32\pi^2}{\gamma^4}. \tag{124}$$

The first two terms $\mathcal{O}\left(1/\gamma^2\right)$ in the local pair correlation function, Eq. (124), were originally derived in Ref. [30] via a second-order perturbative expansion for small parameter $1/\gamma \ll 1$, see Eq. (23). Additional thermal corrections at order $\mathcal{O}\left(1/\gamma^2\right)$ have been reported in Ref. [64]. Here we retain all contributions to order $\mathcal{O}\left(1/\gamma^4\right)$ for a better agreement with the thermal Bethe ansatz exact findings (see Sec. 5).

Thermodynamic properties in this regime VI have been already calculated within the Luttinger Liquid theory [62] and the hard-core prescription without any further series expansion for large interaction strength and low temperatures [63, 64]. However, as discussed above for regime V in subsection 4.3.1, our results make explicit the dependence on $\gamma$ and $\tau$ of dominant contributions to the thermodynamics, while matching well with the thermal Bethe ansatz predictions, see Sec. 5.

In the results for the thermodynamic quantities provided above, the corresponding findings for the ideal Fermi gas in the degenerate regime ($\tau \ll \pi^2$) can be derived by applying the low-temperature Sommerfeld expansion [63, 64, 92]:

$$P_{\text{IFG}} = \frac{\hbar^2 n^3}{2m} \left( \frac{2\pi^2}{3} + \frac{\tau^2}{6} + \frac{\tau^4}{30\pi^2} + \frac{35\tau^6}{648\pi^4} \right), \tag{125}$$

$$S_{\text{IFG}} = Nk_B \left( \frac{\tau}{6} + \frac{\tau^3}{45\pi^2} + \frac{7\tau^5}{216\pi^4} \right), \tag{126}$$

$$\mu_{\text{IFG}} = \frac{\hbar^2 n^2}{2m} \left( \pi^2 + \frac{\tau^2}{12} + \frac{\tau^4}{36\pi^2} + \frac{7\tau^6}{144\pi^4} \right), \tag{127}$$

$$E_{\text{IFG}} = N\frac{\hbar^2 n^2}{2m} \left( \frac{\pi^2}{3} + \frac{\tau^2}{12} + \frac{\tau^4}{60\pi^2} + \frac{35\tau^6}{1296\pi^4} \right), \tag{128}$$

$$C_{L,\text{IFG}} = Nk_B \left( \frac{\tau}{6} + \frac{\tau^3}{15\pi^2} + \frac{35\tau^5}{216\pi^4} \right), \tag{129}$$

$$\kappa_{T,\text{IFG}}^{-1} = \frac{\hbar^2 n^3}{2m} \left( 2\pi^2 - \frac{\tau^2}{6} - \frac{\tau^4}{6\pi^2} - \frac{35\tau^6}{72\pi^4} \right). \tag{130}$$

## 4.4 Application to inhomogeneous trapped systems

Whilst the results derived thus far are applicable only to uniform gases, it is straightforward to generalise them to inhomogeneous (e.g., harmonically trapped) systems within the so-called

local density approximation (LDA) [31, 65, 93]. The LDA prescription holds if the gas contains a large number of atoms $N$ and it is confined in a sufficiently weak external trapping potential $V(x)$, for which the density profile $n(x)$ is slowly varying with the spatial axial coordinate $x$ compared to the characteristic lengthscale of the trapping potential itself. Under such conditions, one can assume that at each position $x$, the local system is approximately homogeneous and is at thermal equilibrium at global temperature $T$ and with the local chemical potential $\mu(x) = \mu_0 - V(x)$, where $\mu_0$ is the global chemical potential of the entire trapped system. The local chemical potential is generally a nonlocal functional of the density $n(x)$, i.e., $\mu(x) = \mu[n(x)]$, whereas the global chemical potential fixes the total number of particles $N$ in the trapped system through the normalization condition, $N = \int_{-\infty}^{\infty} n(x)\, dx$, where the density profile $n(x)$ is calculated from the definition of $\mu(x)$.

The LDA enables then to obtain the thermodynamics of a trapped gas starting from its knowledge in the uniform configuration. As an example, we use the findings of the preceding Sections to calculate the equation of state $\mu[n(x)]$ and thus the density profile $n(x)$ of a harmonically trapped 1D Bose gas in all regimes of Fig. 1, which are defined in terms of the interaction strength and temperature. However, the LDA prescription is very general as it is readily applicable to other shapes of the trapping potentials (different from the harmonic one; see, e.g., [94]).

For inhomogeneous systems, it is convenient to employ a global (density-independent) dimensionless temperature $\mathcal{T}$ [31]:

$$\mathcal{T} = \frac{2\hbar^2 k_B T}{m g^2} = \frac{\tau}{\gamma^2}, \tag{131}$$

where the thermal energy $k_B T$ is rescaled in terms of the characteristic scattering energy $m g^2 / (2\hbar^2)$ and we consider the interaction strength $\gamma$ (3) and temperature $\tau$ (4) defined for the uniform gas. In terms of the global dimensionless parameters $\mathcal{T}$ and $\gamma$, the chemical potential $\mu$ in each of the six regimes of Fig. 1 and reported in Sec. 4 for the uniform 1D Bose system can be expressed as:

$$\text{I}: \quad \frac{\mu}{k_B T} = \frac{1}{\mathcal{T}\gamma^2}\left(2\gamma - \frac{2}{\pi}\gamma^{3/2} + \frac{\pi}{24}\mathcal{T}^2\gamma^{7/2} - \frac{\pi^3}{384}\mathcal{T}^4\gamma^{11/2} + \frac{\pi^5}{1024}\mathcal{T}^6\gamma^{15/2}\right), \tag{132}$$

$$\text{II}: \quad \frac{\mu}{k_B T} = \frac{1}{\mathcal{T}\gamma^2}\left[2\gamma + \frac{1}{2}\mathcal{T}\gamma^{5/2} + \frac{\zeta(1/2)}{\sqrt{\pi}}\mathcal{T}^{1/2}\gamma^2 - \frac{2\zeta(-1/2)}{\sqrt{\pi}}\mathcal{T}^{-1/2}\gamma\right], \tag{133}$$

$$\text{III}: \quad \frac{\mu}{k_B T} = \frac{1}{\mathcal{T}\gamma^2}\left\{-\frac{\pi\mathcal{T}\gamma^2}{\left[\frac{2\sqrt{\pi}}{\gamma\sqrt{\mathcal{T}}} - \zeta(1/2)\right]^2} + 4\gamma - \frac{10}{\mathcal{T}^2\gamma^2}\right\}, \tag{134}$$

$$\text{IV}: \quad \frac{\mu}{k_B T} = \frac{1}{\mathcal{T}\gamma^2}\left[\mathcal{T}\gamma^2\ln\left(\frac{2\sqrt{\pi}}{\gamma\sqrt{\mathcal{T}}}\right) - \gamma\sqrt{2\pi\mathcal{T}}\right.$$
$$\left. + 3\pi\left(1 - \frac{4}{3\sqrt{3}}\right) + 4\pi^{3/2}\frac{(2\sqrt{3}-5)}{3\gamma\sqrt{2\mathcal{T}}} + 4\gamma - \gamma\sqrt{\frac{2\pi}{\mathcal{T}}}\right], \tag{135}$$

$$\text{V}: \quad \frac{\mu}{k_B T} = \frac{1}{\mathcal{T}\gamma^2}\left[\mathcal{T}\gamma^2\ln\left(\frac{2\sqrt{\pi}}{\gamma\sqrt{\mathcal{T}}}\right) + \gamma\sqrt{2\pi\mathcal{T}} + 3\pi\left(1 - \frac{4}{3\sqrt{3}}\right) + 4\pi^{3/2}\frac{(2\sqrt{3}-5)}{3\gamma\sqrt{2\mathcal{T}}}\right.$$
$$\left. - 4\mathcal{T}\gamma - 3\sqrt{2\pi\mathcal{T}} - \frac{8\pi}{\gamma}\left(2 - \frac{8}{3\sqrt{3}}\right) + 6\mathcal{T} + \frac{8\sqrt{2\pi\mathcal{T}}}{\gamma} - \frac{32\mathcal{T}}{3\gamma}\right], \tag{136}$$

$$\text{VI}: \quad \frac{\mu}{k_B T} = \frac{1}{\mathcal{T}\gamma^2}\left(\pi^2 + \frac{\mathcal{T}^2\gamma^4}{12} + \frac{\mathcal{T}^4\gamma^8}{36\pi^2} + \frac{7\mathcal{T}^6\gamma^{12}}{144\pi^4} - \frac{16\pi^2}{3\gamma} + \frac{20\pi^2}{\gamma^2} - \frac{1}{3}\mathcal{T}^2\gamma^2 - \frac{64\pi^2}{\gamma^3}\right).$$
(137)

By fixing the temperature $\mathcal{T}$ of the atomic cloud at thermal equilibrium, the above results provide a relationship between $\mu$ and $\gamma$. We then invoke the LDA for the calculation of the local chemical potential $\mu(x) = \mu_0 - V(x)$, where $\mu_0 = \mu(x=0)$ is the chemical potential at the center of the cloud and $V(x) = \frac{1}{2}m\omega_x^2 x^2$ is the harmonic trapping potential with $\omega_x$ being the axial trap frequency. This allows one to find the relationship between $\mu(x)$ versus $\gamma(x)$ at any $x$. We invert this function graphically to obtain $\gamma(x)$ versus $\mu(x)$, and then the local interaction strength $\gamma(x)$ versus the spatial coordinate $x$. Finally, we recognise that $1/\gamma(x)$ is proportional to $n(x)$ and therefore we have built the numerical density profile $n(x)$ (in certain units) as a function of $x$ [31].

As a crude approximation to this numerical inversion procedure, here we retain only the first term in Eqs. (132) – (137) and solve analytically for the density profile.

In regimes I and II, this yields the well-known Thomas-Fermi (TF) inverted parabola [31, 65, 95, 96],

$$n(x) = n_0(1 - x^2/R_{\text{TF}}^2),$$
(138)

for $|x| < R_{\text{TF}}$, and $n(x) = 0$ otherwise. The central density is $n_0 = n(x=0) = \mu_0/g$, the Thomas-Fermi radius is defined as $R_{\text{TF}} = \sqrt{2\mu_0/(m\omega_x^2)}$ and the global chemical potential $\mu_0 = (9mg^2N^2\omega_x^2/32)^{1/3}$ has been obtained from the normalization condition.

In regime III, we find the density profile [32],

$$n(x) = \sqrt{\frac{mk_B T}{2\pi\hbar^2}}\zeta(1/2) + n_0\frac{\sqrt{|\mu_0|}}{\sqrt{|\mu_0| + \frac{1}{2}m\omega_x^2 x^2}},$$
(139)

which is valid for distances $|x| \le R_T$, delimited by the thermal radius $R_T = \sqrt{2k_B T/(m\omega_x^2)}$. Here, $n_0 = \sqrt{mk_B^2 T^2/(2\hbar^2|\mu_0|)}$ and $|\mu_0| = k_B T e^{-N\hbar\omega_x/(k_B T)}$.

For regimes IV and V, the density profile is well approximated by the Maxwell-Boltzmann distribution describing a classical ideal gas at high temperatures [31],

$$n(x) = n_0 e^{-x^2/R_T^2},$$
(140)

where the peak density $n_0 = N/(\sqrt{\pi}R_T)$ is fixed by the normalization condition and the thermal radius $R_T$ determines the width of the Gaussian. The thermal radius here has the same definition of the one appearing in the density profile for regime III above.

Finally, in regime VI, we employ the theory of the Tonks-Girardeau gas at zero temperature to obtain the Thomas-Fermi semi-circle density profile [31, 96, 97]:

$$n(x) = n_0\sqrt{1 - x^2/R_{\text{TF}}^2},$$
(141)

which is valid only for distances smaller than the Thomas-Fermi radius $R_{\text{TF}}$: $|x| < R_{\text{TF}}$, and $n(x) = 0$ otherwise. The central density is $n_0 = \sqrt{2m\mu_0/(\pi^2\hbar^2)}$ and from the normalization condition of the number of particles, one finds the global chemical potential $\mu_0 = N\hbar\omega_x$. The Thomas-Fermi radius is defined exactly as in Eq. (138).

In Fig. 2, we report the regime diagram in terms of the interaction strength and temperature (to be compared with Fig. 1) for the uniform 1D Bose gas in the $(\gamma, \tau)$ parameter space [panel (a)] and then the same diagram, but now in the $(\gamma, \mathcal{T})$ parameter space [panel (b)], which is more suitable for treating inhomogeneous systems, such as the harmonically trapped

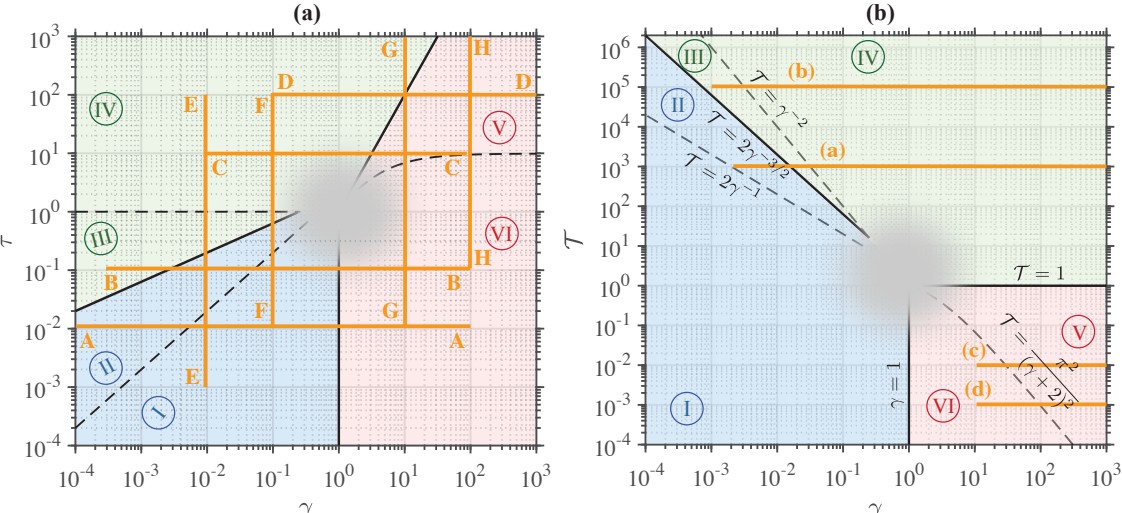

Figure 2: Diagram of the six regimes I–VI for the 1D Bose gas (see Fig. 1) in terms of the interaction strength $\gamma$, Eq. (3), and dimensionless temperature. Panel (a) refers to the uniform configuration where the temperature $\tau$, Eq. (4), depends on the density $n$. Orange lines crossing different regimes correspond to the constant values of $\gamma$ or $\tau$ for which the analytical results for the thermodynamic properties reported in Sec. 4 will be compared with the exact thermal Bethe ansatz findings in Figs. 4–9 below. Panel (b) shows the same diagram of the six regimes I–VI, but in terms of $\gamma$ and the dimensionless temperature $\mathcal{T}$, Eq. (131), which no longer depends on the density $n$ but instead depends on the coupling constant $g$. This is more suitable for describing inhomogeneous (e.g., harmonically trapped) systems in the local density approximation, with a global dimensionless temperature given by $\mathcal{T}$. In this case, the dimensionless interaction strength $\gamma$ can be thought of as representing the local values $\gamma(x) \sim 1/n(x)$ within the inhomogeneous density profile $n(x)$ that now depends on the spatial coordinate $x$ in the LDA. In panel (b), the horizontal orange lines (a), (b), (c), and (d) at four constant values of $\mathcal{T}$ and varying local interaction strength $\gamma(x) \sim 1/n(x)$ correspond to the density profiles $n(x)$ reported in the panels (a), (b), (c), and (d) of Fig. 3, respectively. Here, increasing the values of $\gamma(x)$ (i.e. decreasing $n(x)$) from the leftmost point along these lines is equivalent of moving from the trap center $x = 0$ towards the tails of the density profile located at large $|x|$.

1D Bose gas considered in the present Section. In panel (a), the orange lines at constant interaction strength $\gamma$ (3) and temperature $\tau$ (4) correspond to the set of parameters for which the approximate analytical results for the thermodynamic quantities in each regime I–VI (see Sec. 4) have been compared with the exact thermal Bethe ansatz findings in Figs. 4–9. In panel (b), on the other hand, the horizontal orange lines (a), (b), (c), and (d) at four different values of temperature $\mathcal{T}$ (131) can be interpreted as a spanning of the local interaction strength $\gamma(x)$ starting from the smallest (most-left) values in the trap center $\gamma_0 = \gamma(x = 0) = mg/(\hbar^2 n_0)$ and progressing to the right side by increasing $\gamma(x)$ and the distance $|x|$ away from $x = 0$. As the density profile $n(x)$ is inversely proportional to $\gamma(x)$, the same lines correspond to moving away from the peak value $n_0$ towards the tails of the density distribution, where the density is lower [31]. Lines (a), (b), (c), and (d) represent then $n(x)$ in the panels (a), (b), (c), and (d) of Fig. 3 below, respectively.

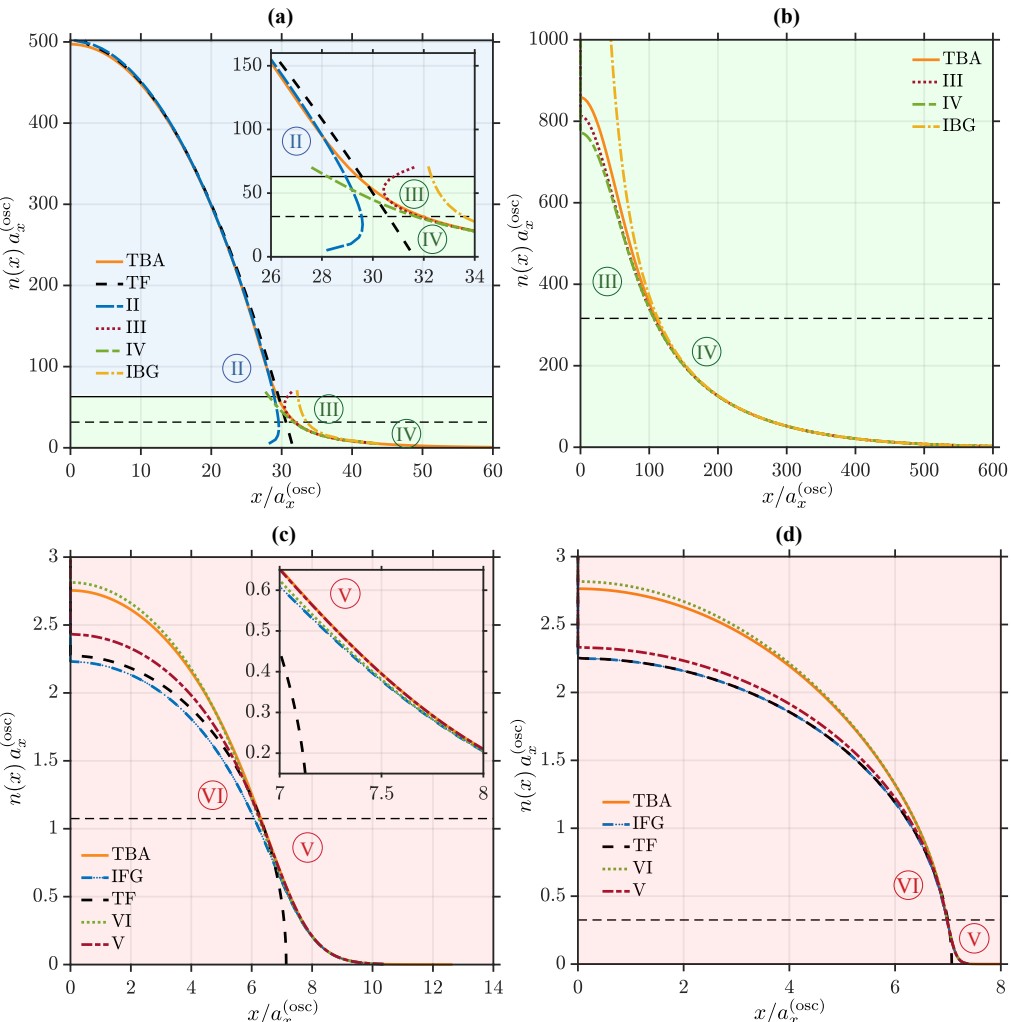

Figure 3: Density profiles $n(x)$ [in units of the harmonic oscillator length $a_x^{(\text{osc})} = \sqrt{\hbar/(m\omega_x)}$] of the harmonically trapped 1D Bose gas at equilibrium. Due to reflectional symmetry about $x = 0$, we plot results for positive spatial coordinate $x > 0$ only. Each panel corresponds to lines (a), (b), (c), and (d) reported in Fig. 2 (b) at constant temperature $\mathcal{T}$, Eq. (131), and changing local interaction strength $\gamma(x)$. In each panel, we compare the exact thermal Bethe ansatz solution (solid orange line) with our approximate analytic results of section 4.4, for a given value of $\mathcal{T}$ and the global dimensionless chemical potential $\mu_0/(Nk_BT)$. In panel (a), the gas spans regimes II and IV, with $\mathcal{T} = 10^3$ and $\mu_0/(k_BT) = 1$, resulting—in the TBA approach—in the dimensionless interaction strength at the trap centre $\gamma_0 = 2 \times 10^{-3}$ and an average number of atoms $N = 2134$. In panel (b), we explore regimes III and IV, with $\mathcal{T} = 10^5$ and $\mu_0/(k_BT) = 10^{-5}$, resulting (in the TBA approach) in $\gamma_0 = 1.2 \times 10^{-3}$ and $N = 196788$. In panel (c), we show findings crossing regimes VI and V, for $\mathcal{T} = 10^{-2}$ and $\mu_0/(k_BT) = 5.1$, resulting in $\gamma_0 = 11.5$ and $N = 30.4$. In panel (d), the system again spans regimes VI and V, but now at lower temperature, $\mathcal{T} = 10^{-3}$ and $\mu_0/(k_BT) = 50$, resulting in $\gamma_0 = 11.5$ and $N = 29.8$. Legends: TBA – thermal Bethe ansatz; II to VI – local density approximation applied to Eqs. (133) – (137); IBG – ideal Bose gas; IFG – ideal Fermi gas (mapped to the strongly-interacting Tonks-Girardeau regime); TF – Thomas-Fermi inverted parabola, Eq. (138), in panel (a), and TF semi-circle, Eq. (141), in panels (c)–(d).

In Fig. 3, we report the density profile $n(x)$ of the harmonically trapped Lieb-Liniger gas at different global temperatures $\mathcal{T} = 10^3, 10^5, 10^{-2}, 10^{-3}$, corresponding to panels (a), (b), (c), and (d), respectively. We compare the findings obtained with different methods: i) exact TBA approach combined with the LDA, ii) numerical inversion applied to Eqs. (132) – (137) within the local density approximation, and iii) simple analytic density profile (138)–(141). The density profiles are shown in nontrivial regimes that cannot be described by the single, simple analytic approximations from Eqs. (138)–(141). Instead, they are better captured by the LDA procedure applied to Eqs. (132)–(137), in a piece-wise way, according to the local regime of the gas. Indeed, the latter results exhibit a better agreement with the TBA solution, differently from the corresponding simple analytic density profiles.

In Fig. 3 (a), none of the individual LDA curves describing the local conditions of the atomic cloud in regimes II, III, or IV, agree with the TBA result across all spatial coordinates $x$. However, they capture the TBA density profile very well if applied piece-wise, in regions belonging (within the local density profile) to regimes II, III, and IV, separately. The only missing regions in such piece-wise fits are near the crossover boundaries between different regimes; however, the inset in Fig. 3 (a) suggests a relatively straightforward recipe for truncating the respective curves at an appropriate position $x$ and extrapolating the missing data in between. The example of Fig. 3 (a) is chosen for a temperature value, for which thermal fluctuations and thermal tails of the density distribution are not negligible so that the entire profile $n(x)$ cannot be well approximated by a single TF inverted parabola, Eq. (138), for all spatial coordinates $x$. The thermal tails at very large $|x|$, where the density is very low and the effect of interactions are negligible, are instead well approximated by the IBG curve from Eq. (87) combined with LDA, whereas at intermediate $|x|$ the wings of the density profile are better described by Eqs. (134) and (135) for regimes III and IV (see also the inset), which include the dependence on the interaction strength $\gamma$.

Similarly, Figs. 3 (b), (c), and (d) illustrate our results in other nontrivial regimes, where the density profiles cannot be described by either a simple Maxwell-Boltzmann distribution characteristic of a high-temperature classical ideal gas (140), or a TF semi-circle of a near-zero temperature Tonks-Girardeau gas, Eq. (141). Instead, these sets of results, which are more likely to be encountered in experiments [12, 56–58, 98–100], lie in intermediate regimes and are better described piece-wise by our LDA density profiles calculated from Eqs. (132)–(137), where we have considered higher-order terms in the interaction strength and temperature. For example, in Fig. 3 (b), the density profile is shown for a higher temperature $\mathcal{T}$ than in (a), and with the value of $\gamma_0$ in the trap centre that belongs to regime III. The TBA density profile is far from either a simple TF parabola (not shown) or a Maxwell-Boltzmann distribution, and instead, is better approximated by our analytic results corresponding to Eqs. (134) and (135). In Figs. 3 (c) and (d), we show examples of density profiles at even lower temperatures, but now with the value of $\gamma_0$ lying in the strongly interacting regime VI. For the TBA density profile in Fig. 3 (c), the central bulk lies in region VI and is better approximated by Eq. (137), whereas the tail, which corresponds to region V, is better captured by Eq. (136). We point out here that obtaining the TF semicircle density profile in the strongly interacting regime, Eq. (141), requires the chemical potential $\mu$ to be well approximated by the zero-temperature result $\mu_{\text{IFG}}(T=0) = \hbar^2 \pi^2 n^2/(2m)$ within the ideal Fermi gas model, see Eq. (127). By comparing $\mu_{\text{IFG}}(T=0)$ with Eq. (137), we find that in order to justify the validity of the TF approximation (141), i.e., that the majority of atoms in the cloud is well contained within the local regime VI, one has to satisfy the conditions $\mathcal{T}\gamma^2 \ll 2\sqrt{3}\pi$ and $\gamma \gg 16/3$. For the two strongly-interacting systems of Fig. 3 (c) and (d), the first condition at the trap center (where $\gamma_0 = 11.5$) is satisfied, whereas the second one is not well fulfilled. As we go further away from the cloud centre at $x = 0$, the local interaction strength $\gamma(x)$ increases and, as a consequence, the second condition becomes well satisfied, but then the first condition breaks down. For this

reason, even though the gas is strongly interacting and at relatively low temperatures, the density profiles cannot be well modelled by a single TF semi-circle, which, indeed, does not show any agreement with the TBA exact findings.

The density profiles reported in Fig. 3, using our analytic results and the piece-wise LDA approach, suggest a novel and relatively simple procedure of curve fitting to experimentally measured *in-situ* density distributions, especially in the cases where obtaining the TBA profiles is computationally more expensive. Such a curve fitting procedure would then be very similar to bi-modal fits of density profiles after time-of-flight expansion, which are routinely used in the experimental measurements in 3D (partially condensed) atomic Bose-Einstein condensates (BECs) [101, 102]. Indeed, in 3D BECs, the utility of bi-modal fits is due to the unambiguous distinction between the condensate mode and the thermal cloud. However, the same approach does not apply to pure 1D Bose systems, which do not undergo a phase transition to a condensate state, but instead are characterised by the smooth crossovers between different regimes. Our new LDA findings, describing the density profiles in each of the regimes, enable such bi-modal or even tri-modal fits, which can in turn be used in the future for simple analytic thermometry in harmonically trapped 1D Bose gases instead of a computationally more involved exact Yang-Yang thermometry [56–58, 99].

Finally, we point out that the same procedure within the LDA can be used to construct all the other thermodynamic quantities of interest for trapped systems. These equilibrium properties are the same as outlined in Sec. 4 for uniform configurations, but now are defined locally as a function of the spatial coordinate $x$. Moreover, for additive quantities, such as the internal energy $E$ or the entropy $S$ of the gas, one can further integrate the respective local results $E(x)$ or $S(x)$ over $x$ and obtain the total internal energy or the total entropy of the trapped system.

# 5 Comparison with the thermal Bethe ansatz exact method

In this Section, we return to the case of a uniform 1D Bose gas. In particular, we compare our analytic results reported in Sec. 4 for the local pair correlation function, $g^{(2)}(0)$, the Helmholtz free energy, $F$, and the chemical potential, $\mu$, with the exact predictions of the thermal Bethe ansatz method. We also discuss the level of quantitative agreement between these analytical and numerical findings.

## 5.1 Local pair correlation function

In Fig. 4, we plot our approximate analytic results for the local pair correlation function, $g^{(2)}(0)$, in six regimes (I–VI) as a function of the interaction strength $\gamma$, Eq. (3). Each panel corresponds to a different temperature $\tau$, Eq. (4): (a) $\tau = 10^{-2}$; (b) $\tau = 10^{-1}$; (c) $\tau = 10$; and (d) $\tau = 10^2$. We compare the analytical approximations with the exact solution obtained using the thermal Bethe ansatz approach (orange solid line). The values of temperature $\tau$ are chosen to expect some disagreement between analytical and TBA results, especially for panels (b) and (c), which are for values of $\tau$ not sufficiently deep into the $\tau \ll 1$ or $\tau \gg 1$ regime required for the approximate analytic results to perform well. Such a disagreement also occurs for interaction strengths $\gamma$ in the vicinity of a boundary between two neighbouring regimes (signalled by vertical dashed and solid lines). Sufficiently far away from such boundaries, where one enters in the deep regimes, the respective analytical results are fully valid and indeed agree extremely well with the exact TBA findings. This is particularly true for very small and very large values of temperature, such as $\tau = 10^{-2}$ and $\tau = 10^2$, in panels (a) and (d), respectively. The agreement further improves for even smaller and larger values of $\tau$ and can be also noticed over broader ranges of the interaction strength $\gamma$ within each region—up to the very close proximity to the regime boundaries. In panel (b), on the other hand, which



Figure 4: Local pair correlation $g^{(2)}(0)$ as a function of the interaction strength $\gamma$ (3) for the uniform 1D Bose gas in different regimes I-VI, see Fig. 1. In each panel, a given value of temperature $\tau$ (4) is reported, corresponding to horizontal orange lines in panel (a) of Fig. 2: (a) $\tau = 10^{-2}$; (b) $\tau = 10^{-1}$; (c) $\tau = 10$; and (d) $\tau = 10^2$. Legends correspond to: TBA – thermal Bethe ansatz; II (2003) – Eq. (16) [30]; I – Eq. (42); II – Eq. (53); III – Eq. (19); IV – Eq. (20); V – Eq. (109); VI – Eq. (124); HC – hard-core from Eq. (100).

is for $\tau = 10^{-1}$ and for regime II, we notice an interesting discrepancy between the exact TBA findings and the approximate analytical result of 2003, Eq. (16), as well as the more accurate new result of the present work, Eq. (53); namely, we observe a large discrepancy between the TBA and these analytic results around $\gamma \simeq 10^{-2}$, even though the validity condition for regime II ($2\gamma \ll \tau \ll 2\sqrt{\gamma}$) is satisfied here. The reason for this discrepancy is that the two analytical predictions are based on the Bogoliubov theory which itself holds only at temperatures strictly lower than the hole-anomaly threshold $\tau < \tau_A$ [64], whereas the temperature of $\tau = 10^{-1}$ in this example is slightly higher than the hole-anomaly temperature $\tau_A$, which is equal to $\tau_A = T_A/T_d \simeq 0.08$, for $\gamma \simeq 10^{-2}$ [64].

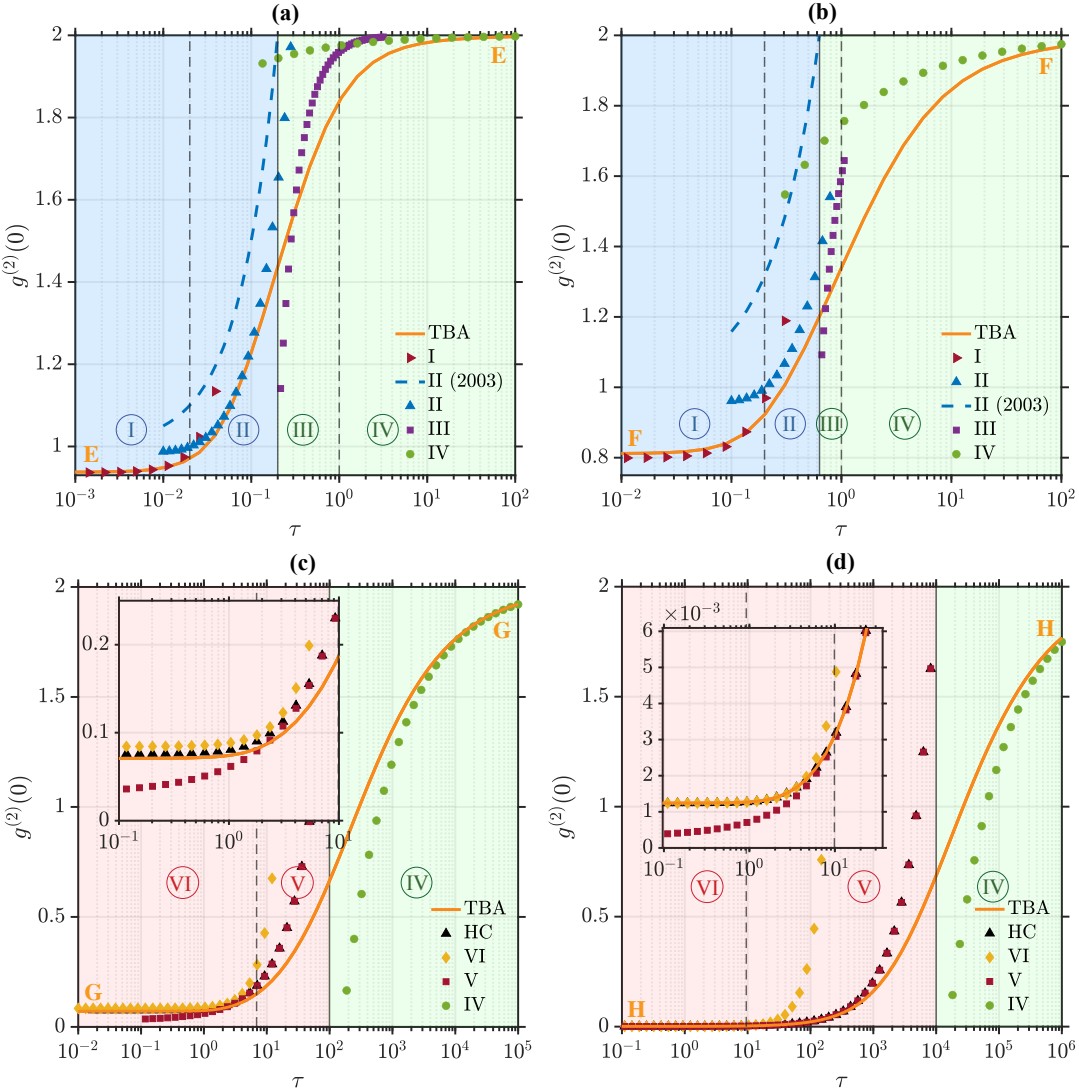

Figure 5: Local pair correlations, similarly to Fig. 4, but here reported as a function of temperature $\tau$. Each panel corresponds to a value of the interaction strength $\gamma$ corresponding to the vertical orange lines in panel (a) of Fig. 2: (a) $\gamma = 10^{-2}$; (b) $\gamma = 10^{-1}$; (c) $\gamma = 10$; and (d) $\gamma = 10^2$.

Fig. 4 also demonstrates that the (crossover) boundaries between different regimes, shown in Fig. 1 and defined via the conditions of applicability of various approximate approaches used to obtain the local pair correlation function $g^{(2)}(0)$ (see Sec. 3), are indeed reasonably accurate. In fact, the analytic curves, plotted in the intended range of $\gamma$ within a certain region, diverge most from the TBA solution near the lower and upper bounds of the respective regime. This can be observed particularly clear in panels (a) and (d), corresponding to the lowest and highest temperature, respectively. Once the regime boundary is crossed, there is a new analytic curve to take over and which provides a better approximation to the TBA exact prediction.

In regimes V and VI in Fig. 4, corresponding to the high- and low-temperature strongly-interacting gas, respectively, we also report the prediction of the hard-core model. In this case, the local pair correlation function (24) has been calculated from the Helmholtz free energy (100) holding for any value of temperature and obtained from the one of the ideal Fermi gas via a "negative excluded volume" replacement of the system size. While the hard-core model data

are valid over the entire strongly-interacting regime, crossing from the ultracold to classical gas, results corresponding to regimes V and VI are high- and low-temperature approximations to the hard-core findings, Eq. (109) and (124), respectively. We find very good agreement between the hard-core results and these approximations, particularly in region V in panels (c) and (d) of Fig. 4, corresponding to higher temperatures. On the other hand, findings for regime VI in panels (a) and (b), reported for lower temperatures, exhibit deviations from the hard-core data, especially for interaction strengths $\gamma$ approaching the boundary with the neighboring regime I.

In panel (a) of Fig. 4, our improved analytic result for the local pair correlation function $g^{(2)}(0)$ in regime II, Eq. (53), shows an excellent agreement with the TBA solution, which is much better than to the corresponding 2003 approximation, Eq. (16) [30] (blue dashed line). The improvement here comes mostly from the third term ($\propto \tau^{1/2}$) in Eq. (53).

In Fig. 5, we present similar results for the local pair correlation function but as a function of temperature $\tau$ for fixed values of the interaction strength $\gamma$ in each panel. All the above physical discussions relative to Fig. 4 can be similarly deduced from Fig. 5. We thus simply present these novel findings as a quantitative record, without any further comment.

The largest deviation of the analytical results from the TBA exact solution is expected for intermediate values of the interaction strength $\gamma \sim 1$ and temperature $\tau \sim 1$, signalled by the blurred grey area in Fig. 1 and where no analytic theory is valid. For this reason, we do not present any of the analytic results for these parameter values.

## 5.2  Helmholtz free energy

In Fig. 6, we present our analytic limits for the absolute value of the Helmholtz free energy per particle $|F|/N$, in regimes I-VI and reported in Sec. 4, as a function of the dimensionless interaction parameter $\gamma$, Eq. (3). A different temperature $\tau$, Eq. (4), is reported in each panel: (a) $\tau = 10^{-2}$; (b) $\tau = 10^{-1}$; (c) $\tau = 10$, and (d) $\tau = 10^2$. As discussed above for the local pair correlation function, we find excellent agreement even for the free energy, between our analytic results and the exact thermal Bethe ansatz solution (orange solid line) over all values of interaction strength and temperature, sufficiently far away from the boundaries separating different regimes. In addition, we show the predictions of the ideal Bose gas ($\gamma = 0$), Eq. (60), and the ideal Fermi gas ($\gamma \to +\infty$), Eq. (97), models, both valid for any $\tau$. In the weak coupling limit, as $\gamma \to 0$, our approximate analytic expressions for finite $\gamma$ and for various regimes, asymptotically approach the IBG prediction, as expected. Similarly, close to the Tonks-Girardeau limit with $\gamma \to \infty$, our approximations reproduce the IFG result. Away from these strict $\gamma = 0$ and $\gamma \to \infty$ limits, the analytic findings at finite $\gamma$ show a much better agreement with the TBA solution than the pure IBG and IFG predictions do. Finally, we also present the result of the hard-core model, Eq. (100), holding for large but finite $\gamma$ and for any $\tau$. We notice that the hard-core model shows an excellent agreement with both the analytical limits at high-temperature (regime V) and low-temperature (regime VI) for strong interactions.

The sharp apparent minima in the thermal quasicondensate regime II of Figs. 6 (a)-(b) are the artefacts of the logarithmic scale of the plots and are due to a change of sign in the free energy $F$ for some values of the interaction strength $\gamma$. $F$ itself is a monotonic function of $\gamma$. In Figs. 6(c) and 6(d), where only high-temperature regimes are presented, the free energy is instead negative $F < 0$ for all the values of $\gamma$ and hence no such behaviour of change of sign is visible. This zero crossing of the free energy is observed for any temperature $\tau < 10$, where the value of $\tau$, corresponding to $F = 0$, is a monotonically increasing function of $\gamma$. By considering the low- and high-temperature limits of the free energy in the ideal Bose and Fermi gas limits, Eqs. (70), (116), (84), and (101), we notice that while $F_{\text{IBG}} < 0$ for any $\tau$, $F_{\text{IFG}} > 0$ for sufficiently small $\tau$ and becomes negative by raising the temperature. This implies that for some finite values of the interaction strength $0 < \gamma < +\infty$, the free energy must vanish.

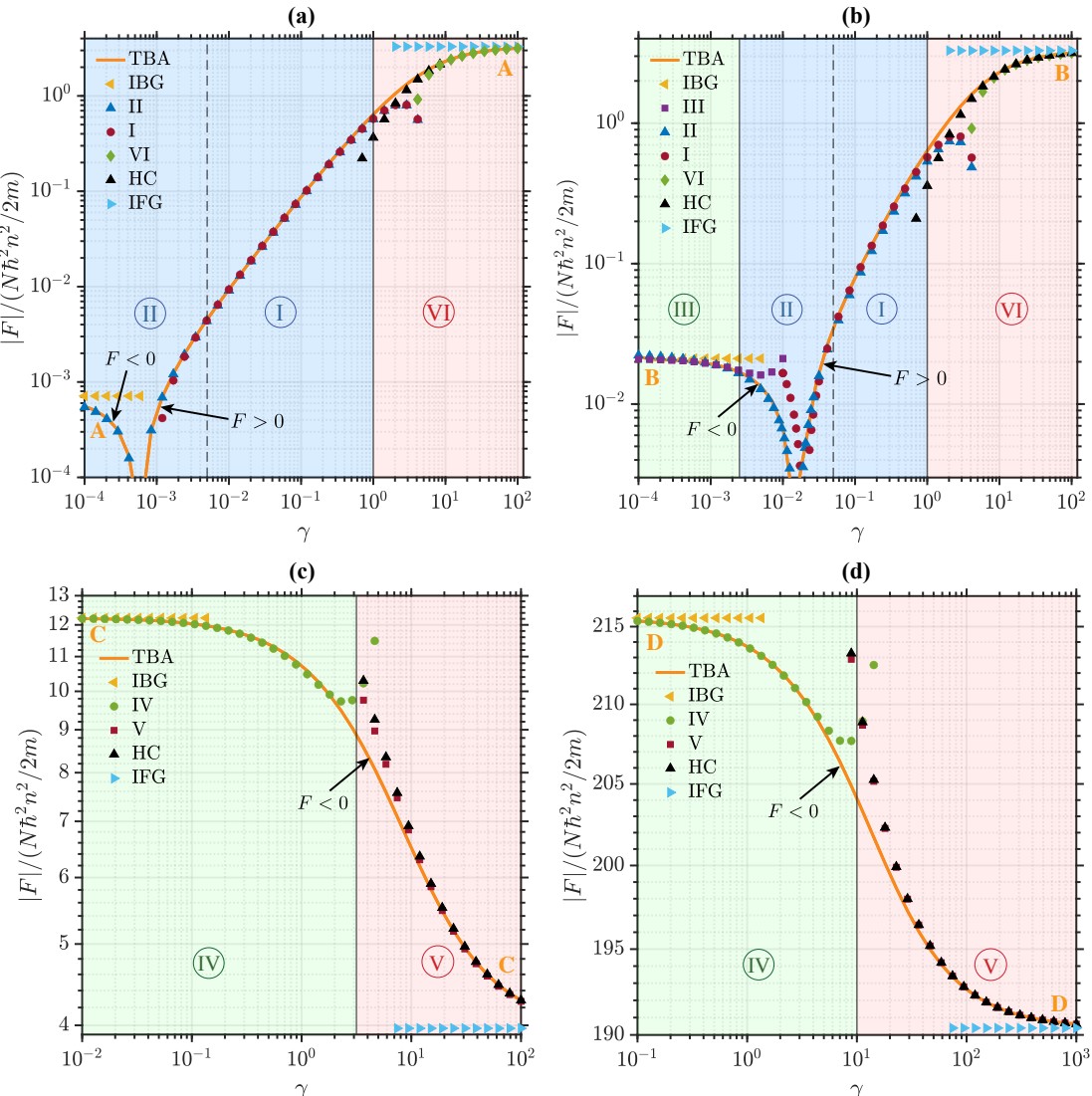

Figure 6: Absolute value of the Helmholtz free energy per particle $|F|/N$ as a function of the interaction strength $\gamma$, Eq. (3), for the uniform 1D Bose gas in different regimes I-VI, see Fig. 1. In each panel, a given value of temperature $\tau$, Eq. (4), is reported, corresponding to the horizontal orange lines in panel (a) of Fig. 2: (a) $\tau = 10^{-2}$; (b) $\tau = 10^{-1}$; (c) $\tau = 10$; and (d) $\tau = 10^2$. Legends correspond to: TBA – thermal Bethe ansatz; I – Eq. (35); II – Eq. (44); III – Eq. (62); IV – Eq. (77); V – Eq. (102); VI – Eq. (117); IBG – ideal Bose gas, Eq. (60); IFG – ideal Fermi gas, Eq. (97); HC – hard-core, Eq. (100).

In Fig. 7, we plot the findings for the Helmholtz free energy, but as a function of the temperature $\tau$ for a given value of the interaction strength $\gamma$ in each panel: (a) $\gamma = 10^{-2}$; (b) $\gamma = 10^{-1}$; (c) $\gamma = 10$; and (d) $\gamma = 10^2$. We arrive at conclusions similar to the ones discussed for Fig. 6. In all panels, the free energy changes sign as we scan $\tau$. In Figs. 7(c) and 7(d), we notice that the analytical limit for the free energy in regime IV, Eq. (77), agrees well with the TBA solution even well into regime V. Similarly, the applicability of the approximate result for regime V, Eq. (102), extends well into regime IV. In the quantum quasicondensate regime I of panels Fig. 7(a) and (b), the Helmholtz free energy is nearly independent of the temperature as the respective curve is almost flat due to the fact that thermal fluctuations

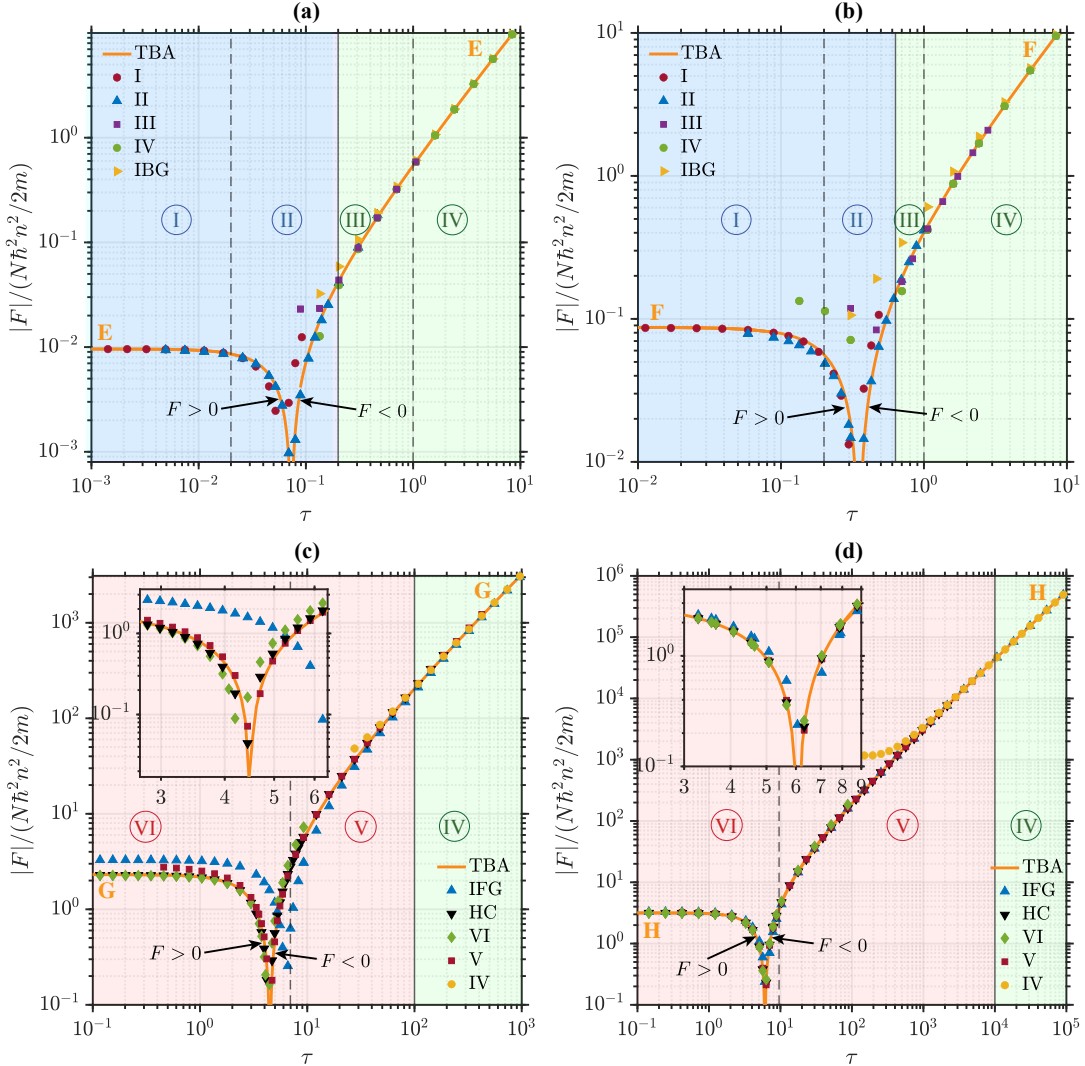

Figure 7: Absolute value of the Helmholtz free energy per particle, similarly to Fig. 6, but here reported as a function of temperature $\tau$. Each panel corresponds to a value of the interaction strength $\gamma$ corresponding to vertical orange lines in panel (a) of Fig. 2: (a) $\gamma = 10^{-2}$; (b) $\gamma = 10^{-1}$; (c) $\gamma = 10$; and (d) $\gamma = 10^2$.

are negligible in this regime. However, in the thermal quasicondensate regime II, the free energy varies significantly with temperature and such a strong thermal dependence is also reproduced in other thermodynamic properties including the local pair correlation function and the chemical potential (see below). Therefore, additional finite-temperature ($\tau \neq 0$) terms in the thermodynamics prove to be not necessarily negligible in regime II, as already discussed above for the local pair correlation function in section 5.1. This highlights one of the key findings of the present work.

We emphasize here again that the free energy in regime II has not been previously described analytically. Our new result agrees well with the exact TBA solution and gives a significant improvement over the previously known analytic approximation from the neighbouring regime I, extrapolated into regime II, as can be seen in panels (a) and (b) of Figs. 6-7.

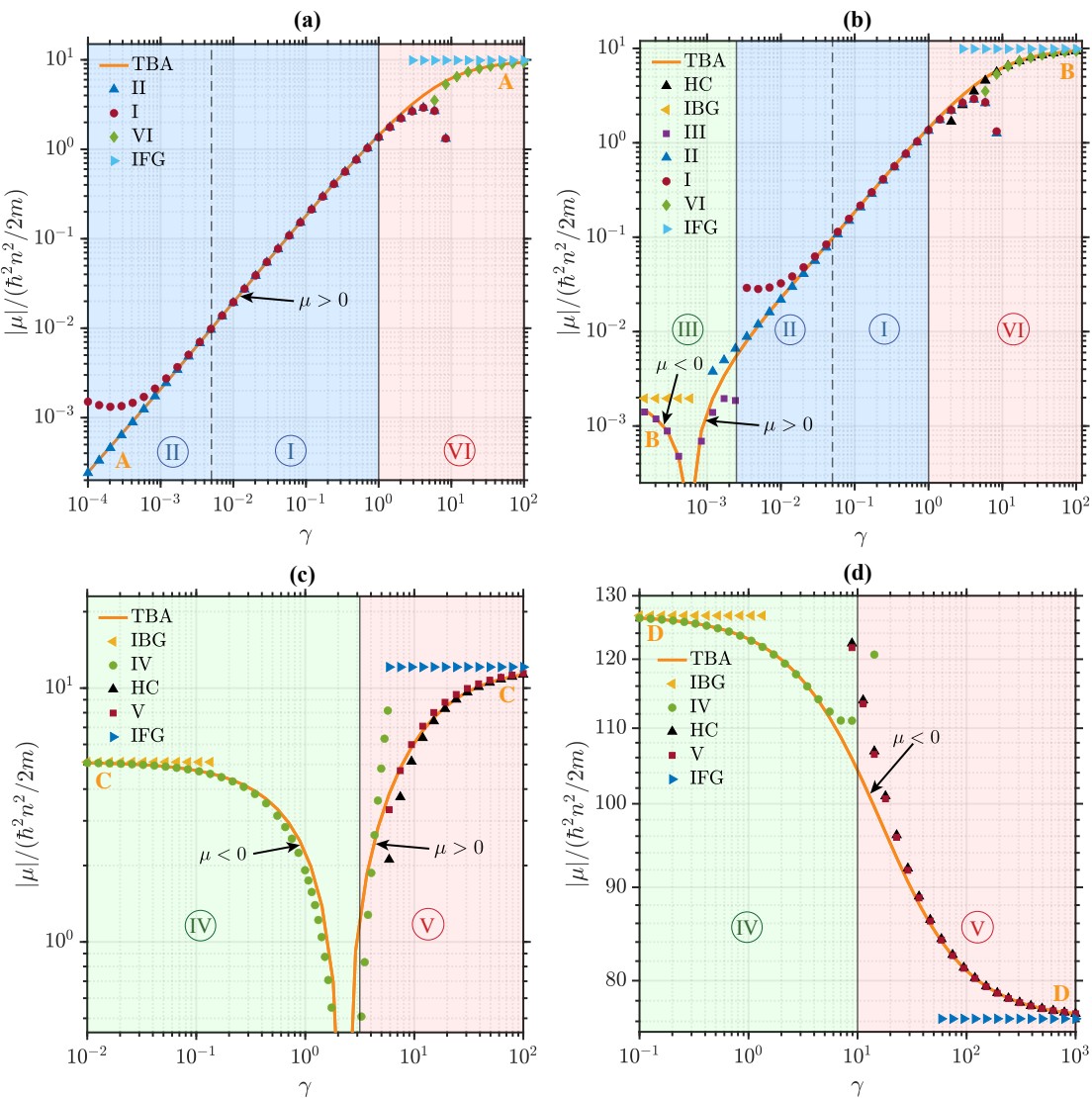

Figure 8: Absolute value of the chemical potential $|\mu|$ as a function of the interaction strength $\gamma$, Eq. (3), for the uniform 1D Bose gas in different regimes I-VI (see Fig. 1). The four panels correspond, respectively, to four fixed values of the temperature parameter $\tau$, Eq. (4), corresponding to horizontal orange lines in panel (a) of Fig. 2: (a) $\tau = 10^{-2}$; (b) $\tau = 10^{-1}$; (c) $\tau = 10$; and (d) $\tau = 10^2$. Legends correspond to: TBA – thermal Bethe ansatz; I – Eq. (38); II – Eq. (49); III – Eq. (65); IV – Eq. (80); V – Eq. (105); VI – Eq. (120); IBG – ideal Bose gas, Eq. (59); IFG – ideal Fermi gas, Eq. (96); HC – hard-core model from Eq. (100).

## 5.3 Chemical Potential

In Fig. 8, we show our analytic results for the absolute value of the chemical potential $|\mu|$, in regimes I-VI, as a function of the interaction strength $\gamma$, Eq. (3), for four different values of temperatures $\tau$, Eq. (4): (a) $\tau = 10^{-2}$; (b) $\tau = 10^{-1}$; (c) $\tau = 10$; and (d) $\tau = 10^2$. As before, we also plot the predictions for the ideal Bose gas, Eq. (59), ideal Fermi gas, Eq. (96), and hard-core model where the chemical potential has been calculated from the corresponding free energy $F$, Eq. (100), via $\mu = (\partial F / \partial N)_{T,L,g}$. In the limit $\gamma \to 0$, our analytic limits for the different regimes reproduce the IBG prediction, whilst for $\gamma \to \infty$ they approach the IFG solution, as expected.

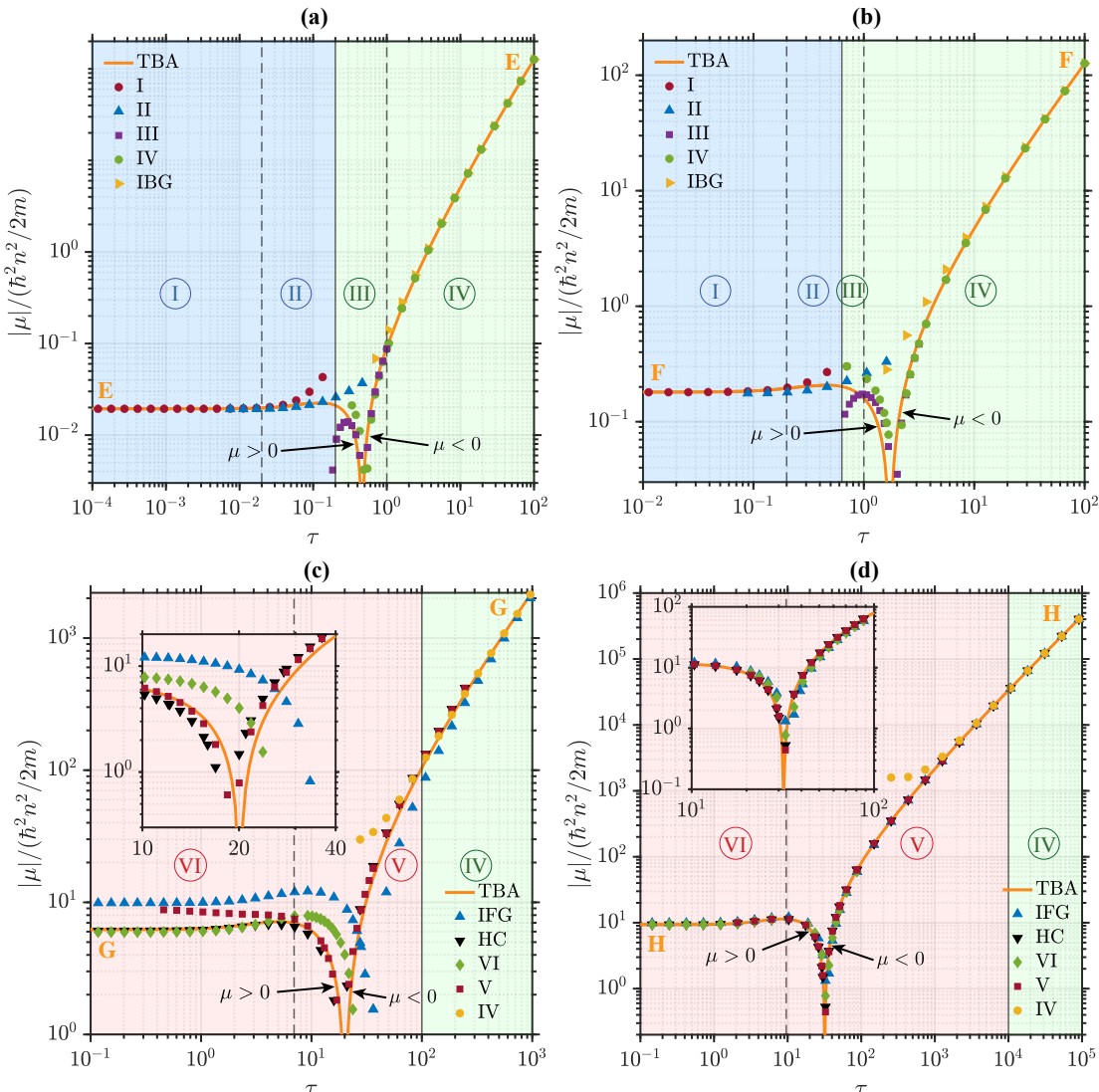

Figure 9: Absolute value of the chemical potential, similarly to Fig. 8, but here reported as a function of temperature $\tau$. Each panel corresponds to a value of the interaction strength $\gamma$, corresponding to vertical orange lines in panel (a) of Fig. 2: (a) $\gamma = 10^{-2}$; (b) $\gamma = 10^{-1}$; (c) $\gamma = 10$; and (d) $\gamma = 10^2$.

The overall picture of the agreement between all these results and the exact thermal Bethe ansatz solution (orange solid line) is similar to that of the Helmholtz free energy, discussed above in section 5.2. We note, in particular, that our new analytic prediction for the chemical potential in regime II (49) shows an excellent agreement with the TBA prediction over a broad range of $\gamma$ within the respective range of validity, as can be seen from Figs. 8 (a) and 8 (b).

The same conclusions are valid when analysing the results for the chemical potential as a function of the dimensionless temperature $\tau$, shown in Fig. 9, for four fixed interaction strengths in each panel: (a) $\gamma = 10^{-2}$; (b) $\gamma = 10^{-1}$; (c) $\gamma = 10$; and (d) $\gamma = 10^2$. For the IBG system with $\gamma = 0$, the chemical potential is always negative and is a monotonically decreasing function of temperature. For any non-zero value of the interaction strength, crossing from the weakly-interacting $\gamma \ll 1$ to the Tonks-Girardeau limit $\gamma \to \infty$ (the latter reproducing the IFG thermodynamics), the chemical potential exhibits a nonmonotonic temperature dependence: it

is a positive increasing function at low temperatures due to phononic excitations [62] and a negative decreasing function at sufficiently high temperatures when approaching the limit of the Maxwell-Boltzmann classical gas. Such a non-monotonic behavior and hence a maximum of the chemical potential as a function of temperature for an arbitrary finite value of the interaction strength $\gamma$ is another manifestation of the hole anomaly [64].

Finally, we notice that in the temperature dependence of the free energy, Fig. 7, and the chemical potential, Fig. 9, the analytical result for regime IV exhibits an agreement with the exact TBA solution even in regime V for temperatures much higher than the hole-anomaly threshold, see panels (c) and (d). The same observation is true even for the analytical prediction of regime V which also holds in regime IV. Both regimes IV and V are non-degenerate and describe the system approaching the ideal Bose and Fermi gas limits, respectively, see Secs. 4.2.2 and 4.3.1. At very high temperatures (well above the hole-anomaly), the effects due to the interaction and the quantum statistics become both negligible and the two regimes exhibit a similar behavior for the various thermodynamic properties. However, the same does not occur for the local pair correlation function $g^{(2)}(0)$ [see panels (c) and (d) of Fig. 5] as in the limit of very high temperatures, $g^{(2)}(0) = 2$ and $g^{(2)}(0) = 0$ in regimes IV and V, respectively, see Eqs. (20) and (22).

## 6 Conclusions

In this work, we have presented a review of analytic treatments of different thermodynamic properties of the homogeneous 1D Bose gas with contact repulsive interatomic interactions, described by the integrable Lieb-Liniger model. The analytic approximations cover six different regimes of the 1D Bose gas identified in Fig. 1 and defined in terms of two dimensionless parameters—the interaction strength $\gamma$, Eq. (3), and the temperature $\tau$, Eq. (4). The utility of these approximations is their analytic transparency and physical insight, compared to the exact but pure numerical approach facilitated by the thermal Bethe ansatz method.

In three of the regimes of the 1D Bose gas (I, V, and VI), our results agree with those reported in the literature previously [30, 62–64]. In regimes III and IV, we found new next-order terms in $\gamma$ in the perturbative treatment of the gas in the nearly ideal Bose gas regime at finite temperatures, compared to the known leading order terms obtained previously using the Hartree-Fock approximation [63, 64] where the dependence on $\gamma$ only appears in the zero-temperature thermodynamics. Finally, in regime II, corresponding to a quasicondensate dominated by thermal (rather than quantum) fluctuations, the analytic approximations developed here have not been previously reported in the literature, to the best of our knowledge, and as such they fill in this important gap.

In addition, we have refined the conditions of applicability of the underlying approximations in each of these six regimes, which in turn allowed us to redefine the crossover boundaries between the different regimes compared to those identified in Ref. [30] in 2003. In all these regimes, our approximate analytic results for the local atom-atom pair correlation function $g^{(2)}(0)$, the Helmholtz free energy $F$, and the chemical potential $\mu$—as function of $\gamma$ and $\tau$—have been compared to the exact TBA results, showing excellent agreement within the respective ranges of applicability of the analytic results and away from the hole-anomaly.

Finally, we have applied our results to treat a harmonically trapped (inhomogeneous) 1D Bose gas using the local density approximation and constructed piece-wise analytic fits to the density profiles evaluated numerically using TBA. These fits are similar to the bi-modal fits to experimentally measured density profiles of 3D partially condensed Bose gases and offer a new, simple method of thermometry of 1D Bose gases, wherein an experimentally measured in-situ density profile can be fitted by a combination of our analytic results in different local

regimes of the trapped system. We also point out that the same procedure within the LDA can be used to construct the other thermodynamic quantities of interest for trapped systems, in addition to enabling the calculation of additive properties, such as the total internal energy or entropy, by simply integrating the respective local results.

Experimentally, various thermodynamic quantities derived here (e. g. , the Helmholtz free energy, the pressure, the entropy, the chemical potential, the internal energy, the heat capacity at constant volume, and the isothermal compressibility) have been extensively measured in 2D and 3D ultracold quantum gases by combining well-calibrated trapping potentials with in-situ measurements of density profiles [55, 103–107]. In 1D Bose gases, this experimental technique has been employed to measure the chemical potential, entropy, pressure, and heat capacity as a function of the interaction strength and temperature [56–58], with the results showing an excellent agreement with the thermal Bethe ansatz predictions. An alternative experimental technique for extracting the thermodynamic properties in ultracold quantum gases, which is preferable in situations when the confining potentials are not well known, is based on the extraction of the isothermal compressibility from the measured atom number fluctuations in individual pixels of in-situ density profiles [12, 100, 108–112]. Such measurements in 1D Bose gases [100, 111, 112] have again shown excellent agreement with the TBA calculations. Finally, we note that the thermodynamic properties can also be accessed from the direct measurement of the local pair correlation function via photoassociation [18, 89] and from the tails of the momentum distribution [113, 114], which is proportional to Tan's contact, using Bragg [115–117] or radio-frequency [118–120] spectroscopy. We hope that all these experimental techniques for characterising the thermodynamic properties of 1D Bose gases may benefit in the future from our analytic results as the first, simplest approximation, before deploying the full machinery of the TBA for comparison with the experimental measurements, which is computationally more involved and hence is much slower.

We remind the reader that early experiments on 1D Bose gases were typically performed by creating an array of multiple 1D Bose gases using 2D optical lattice potentials [14, 16–18, 56, 121–124], or by trapping a single atomic cloud using an optical dipole trap [69] or a magnetic trap created by an atom chip [12, 66, 99, 125] (for a review, see Ref. [11]). The latter two techniques have the great advantage of avoiding the ensemble averaging over many non-identical copies of the system taking place in 2D lattice potentials, in addition to the possibility of exploring a wide range of temperatures, while still maintaining the 1D condition $k_B T, \mu \ll \hbar \omega_\perp$ [126]. On the other hand, 2D optical lattice potentials allow for extremely tight transverse harmonic confinement of the gas, with the transverse trap frequency $\omega_\perp$ reaching tens of kHz, which increases the effective 1D coupling strength $g \simeq 2\hbar\omega_\perp a_{3D}$ and allows one to reach and explore the strongly interacting (Tonks-Girardeau) regime. The strongly interacting regime can also be reached by employing a magnetic Fano–Feshbach resonance [20, 123] to tune the 3D scattering length $a_{3D}$ and hence the 1D coupling strength $g$, or by using a confinement-induced resonances for tuning the 1D scattering length [19, 22]. In all these cases, however, the longitudinal confinement along the axial direction is typically harmonic, which hinders the direct applicability of our results for uniform 1D Bose gases and necessitates the use of the local density approximation. The current generation of 1D Bose gas experiments, on the other hand, exploits state-of-the-art trapping techniques to flatten out the longitudinal confinement [58, 127], which makes the uniform Bose gas results more relevant and more directly applicable. Even ring trap geometries that are not far from approaching the 1D regime are being created nowadays [128, 129]; once in the 1D configuration, such ring traps would make an ideal realization of the uniform Lieb-Liniger gas with periodic boundary condition.

Looking further ahead, our results regarding the equilibrium thermodynamics in a uniform system may play an important role in the revised versions of hydrodynamic theories of 1D Bose gases and the question of collective oscillations, such as breathing modes, in

harmonic traps [15, 21, 130–137]. The frequencies of breathing oscillations, predicted from hydrodynamic descriptions, depend strongly on the underlying thermodynamic equation of state for the pressure of the gas; indeed, such an equation of state enters directly into one of the hydrodynamic equations, and hence its particular form strongly affects their overall solution. Another open question in this area concerns the temperature-induced transition from hydrodynamic to collisionless behaviour predicted to occur for the dipole compression mode in a harmonically trapped 1D Bose gas at both weak and strong interactions [138]. Our improved and new results for various thermodynamic quantities of the 1D Bose gas may help to shed light on this important question, in addition to providing additional insights into the question of thermalisation [139–143] or its apparent lack in 1D Bose gases [122]. Equilibrium thermodynamic properties are also relevant to the behaviour of microscopic quantities, such as the nonlocal pair correlation function $g^{(2)}(r)$ and the momentum distribution of the 1D Bose gas [40, 53, 54, 59, 144]. In particular, the Tan's relation, proportional to the contact parameter and describing the tails of the momentum distribution, has recently been predicted to fade away above the hole-anomaly temperature for any interaction strength [54], and it was shown that such fading is induced by a dramatic thermal increase of the internal energy of the gas. More generally, our improved analytic predictions for the local pair correlation $g^{(2)}(0)$ may provide additional insights into the studies of 1D Bose gases through Tan's thermodynamic relations involving the said contact parameter, which is simply linearly dependent on $g^{(2)}(0)$, Eq. (27), and therefore both are thermodynamic quantities.

Finally, we anticipate that the analytic methods developed and used here can be extended and applied to the study of thermodynamic properties of other related ultracold quantum systems, such as impurities immersed in a quantum fluid of different nature [145–148], the Yang-Gaudin model describing a spin-1/2 Fermi gas in 1D [149,150], and multicomponent systems [151–154]. Other future research avenues include a large variety of quantum liquids; for example, our results may be relevant to 1D liquids of $^4$He, which have been recently realized in different interaction and temperature regimes [155], and to ultracold and ultradilute 1D liquids in binary bosonic mixtures at finite temperature [156], which can be observed in current state-of-the-art experiments [157]. Ultracold and ultradilute quantum liquids emerge due to the effects of quantum fluctuations [158], which are strongly enhanced in 1D and hence provide a greater stability to these systems [159,160] compared to their 3D counterparts. This makes this novel phase of quantum matter an exceptional platform for the investigation of quantum many-body physics.

## Acknowledgements

K. V. K. acknowledges stimulating discussions with D. M. Gangardt.

**Author contributions**  M. L. K., G. D. R., and K. V. K. devised the initial concepts and theory. M. L. K. derived analytical results and performed exact Bethe-ansatz calculations. The bulk of the manuscript was written by M. L. K. and G. D. R., with suggestions and modifications from K. V. K. The project was supervised by G. D. R. and K. V. K.

**Funding information**  K. V. K. acknowledges support by the Australian Research Council Discovery Project Grant No. DP190101515. G. D. R. received funding from the grant IJC2020-043542-I funded by MCIN/AEI/10.13039/501100011033 and by "European Union NextGenerationEU/PRTR". G. D. R. was also partially supported by the grant PID2020-113565GB-C21 funded by MCIN/AEI/10.13039/501100011033 and the grant 2021 SGR 01411 from the Generalitat de Catalunya.

## A  Low-temperature expansion in the quantum quasicondensate regime (I)

In this Appendix, we demonstrate the Taylor series of Eq. (34) which we have employed for the calculation of the thermodynamic properties in the quantum quasicondensate regime (I).

We first note that we can express the quantity $\sqrt{a + ib}$ in terms of real and imaginary parts

$$\sqrt{a+ib} = \sqrt{\frac{a + \sqrt{a^2 + b^2}}{2}} + i\sqrt{\frac{-a + \sqrt{a^2 + b^2}}{2}}. \tag{A.1}$$

It then suffices to consider the imaginary part of $\sqrt{a + ib}$ with $a = 1$ and $b = x$ as appearing in Eq. (34). From the generalised binomial theorem, we can write

$$\sqrt{1 + ix} = \sum_{j=0}^{\infty} \binom{1/2}{j} (ix)^j, \tag{A.2}$$

where $\binom{\alpha}{\beta}$ is the binomial coefficient. To find the imaginary component of the above expression, notice that $i^{2j} = (-1)^j$ and $i^{2j+1} = (-1)^j i$. Hence, we only retain terms in the sum with $j$ odd. This leads to the result

$$\left(\sqrt{1 + x^2} - 1\right)^{1/2} = \sqrt{2}\,\mathrm{Im}\left(\sqrt{1 + ix}\right) = \sqrt{2}\sum_{j=0}^{\infty} \binom{1/2}{2j+1}(-1)^j x^{2j+1}. \tag{A.3}$$

We now make use of the identity

$$\binom{1/2}{j} = \binom{2j}{j}\frac{(-1)^{j+1}}{2^{2j}(2j-1)}, \tag{A.4}$$

to write

$$\begin{aligned}
\left(\sqrt{1 + x^2} - 1\right)^{1/2} &= \sqrt{2}\sum_{j=0}^{\infty} \binom{4j+2}{2j+1}\frac{(-1)^j}{2^{4j+2}(4j+1)}x^{2j+1} \\
&= \sqrt{2}\sum_{j=0}^{\infty} \frac{(4j+2)!}{(2j+1)!(2j+1)!}\frac{(-1)^j}{2^{4j+2}(4j+1)}x^{2j+1} \\
&= \frac{x}{\sqrt{2}} + \sum_{j=1}^{\infty} \frac{(4j-1)!}{(2j+1)!(2j-1)!}\frac{(-1)^j}{2^{4j-1/2}}x^{2j+1},
\end{aligned} \tag{A.5}$$

as desired, i.e., as stated in Eq. (34). It is straightforward to show that the series converges for $0 \le x \le 1$.

## B  High-temperature approximation in the thermal quasicondensate regime (II)

In this Appendix, we justify the approximation (43), which we have used to calculate the Helmholtz free energy in the thermal quasicondensate regime (II), corresponding to $2\gamma \ll \tau \ll 2\sqrt{\gamma}$.

To this end, consider the difference between the following two integrals:

$$
\int_0^\infty du\, \frac{\left(\sqrt{\tau^2 u^2/(4\gamma^2)+1}-1\right)^{1/2}}{e^u-1} - \int_{2\gamma/\tau}^\infty du\, \frac{\sqrt{\tau u/(2\gamma)-1}}{e^u-1} \tag{B.1}
$$

$$
= \int_0^{2\gamma/\tau} du\, \frac{\left(\sqrt{\tau^2 u^2/(4\gamma^2)+1}-1\right)^{1/2}}{e^u-1} + \int_{2\gamma/\tau}^\infty du\, \left[\frac{\left(\sqrt{\tau^2 u^2/(4\gamma^2)+1}-1\right)^{1/2}}{e^u-1} - \frac{\sqrt{\tau u/(2\gamma)-1}}{e^u-1}\right].
$$

The upper limit in the first integral in Eq. (B.1) is small, $2\gamma/\tau \ll 1$, as required by the validity of regime II, Fig. 1. Therefore, we can employ the following approximation for $u \ll 1$,

$$
\frac{1}{e^u-1} \approx \frac{1}{u} - \frac{1}{2} + \mathcal{O}(u), \tag{B.2}
$$

from which we obtain, for the first integral:

$$
\int_0^{2\gamma/\tau} du\, \frac{\left(\sqrt{\tau^2 u^2/(4\gamma^2)+1}-1\right)^{1/2}}{e^u-1} \approx \int_0^{2\gamma/\tau} du\, \frac{\left(\sqrt{\tau^2 u^2/(4\gamma^2)+1}-1\right)^{1/2}}{u}
$$
$$
- \frac{1}{2}\int_0^{2\gamma/\tau} du\, \left(\sqrt{\tau^2 u^2/(4\gamma^2)+1}-1\right)^{1/2}. \tag{B.3}
$$

By implementing the substitution $y = \left(\sqrt{\tau^2 u^2/(4\gamma^2)+1}-1\right)^{1/2}$, we then find

$$
\int_0^{2\gamma/\tau} du\, \frac{\left(\sqrt{\tau^2 u^2/(4\gamma^2)+1}-1\right)^{1/2}}{u} = 2\sqrt{\sqrt{2}-1} - \sqrt{2}\tan^{-1}\left(\frac{\sqrt{\sqrt{2}-1}}{\sqrt{2}}\right), \tag{B.4}
$$

$$
\int_0^{2\gamma/\tau} du\, \left(\sqrt{\tau^2 u^2/(4\gamma^2)+1}-1\right)^{1/2} = \frac{4\gamma\sqrt{2}}{3\tau}\left(1-\sqrt{\sqrt{2}-1}\right). \tag{B.5}
$$

On the other hand, the same series expansion (B.2), cannot be in principle used in the entire range of the second integral of Eq. (B.1), as instead it also includes large values for the variable $u \gg 1$. However, for $u \gg 1$, the exponential functions $e^u$ in the denominators dominate and since such an integral is expressed as a *difference* of integrals, its main contribution comes from $u \ll 1$. We can then use the following approximation, relying on Eq. (B.2):

$$
\int_{2\gamma/\tau}^\infty du\, \left[\frac{\left(\sqrt{\tau^2 u^2/(4\gamma^2)+1}-1\right)^{1/2}}{e^u-1} - \frac{\sqrt{\tau u/(2\gamma)-1}}{e^u-1}\right] \approx
$$
$$
\int_{2\gamma/\tau}^\infty du\, \left[\frac{\left(\sqrt{\tau^2 u^2/(4\gamma^2)+1}-1\right)^{1/2}}{u} - \frac{\sqrt{\tau u/(2\gamma)-1}}{u}\right]
$$
$$
- \frac{1}{2}\int_{2\gamma/\tau}^\infty du\, \left[\left(\sqrt{\tau^2 u^2/(4\gamma^2)+1}-1\right)^{1/2} - \sqrt{\tau u/(2\gamma)-1}\right]. \tag{B.6}
$$

We note here that the second integral is convergent despite each term diverging; the difference of these terms is finite. We calculate

$$
\int_{2\gamma/\tau}^\infty du\, \left[\frac{\left(\sqrt{\tau^2 u^2/(4\gamma^2)+1}-1\right)^{1/2}}{u} - \frac{\sqrt{\tau u/(2\gamma)-1}}{u}\right] =
$$

$$\pi\left(1-\frac{1}{\sqrt{2}}\right)-2\sqrt{\sqrt{2}-1}+\sqrt{2}\tan^{-1}\left(\frac{\sqrt{\sqrt{2}-1}}{\sqrt{2}}\right), \tag{B.7}$$

and

$$\int_{2\gamma/\tau}^{\infty}du\left[\left(\sqrt{\tau^2u^2/(4\gamma^2)+1}-1\right)^{1/2}-\sqrt{\tau u/(2\gamma)-1}\right]=\frac{4\sqrt{2}}{3}\frac{\gamma}{\tau}\sqrt{\sqrt{2}-1}. \tag{B.8}$$

Combining our above results, Eq. (B.1) provides the correction in terms of the interaction strength $\gamma$ and temperature $\tau$,

$$\int_{0}^{\infty}du\,\frac{\left(\sqrt{\tau^2u^2/(4\gamma^2)+1}-1\right)^{1/2}}{e^u-1}-\int_{2\gamma/\tau}^{\infty}du\,\frac{\sqrt{\tau u/(2\gamma)-1}}{e^u-1}\approx\pi\left(1-\frac{1}{\sqrt{2}}\right)-\frac{2\sqrt{2}\gamma}{3\tau}, \tag{B.9}$$

which is precisely Eq. (43).

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
