# Peer review of "Analytic thermodynamic properties of the Lieb-Liniger gas"

_SciPost Physics Core, doi:SciPost Phys. Core 7, 047 (2024)_

## Round 2 · Referee Report · Anonymous (Referee 1) · 2024-5-9

Strengths

1. Comprehensive review of the approximate analytic results for the thermodynamic quantities of the Lieb-Liniger model.
2. New and improved expansions in the thermal quasicondensate regime (regime II), the degenerate nearly ideal Bose gas regime (regime III) and the non-degenerate nearly ideal Bose gas regime (regime IV).

Report

The submitted paper provides a thorough review of the approximate analytic results of the relevant thermodynamic quantities (pressure, entropy, chemical potential, energy, specific heat, isothermal compressibility, and pair correlation function) for the repulsive Lieb-Liniger (LL) model. While in principle all the thermodynamic quantities of the Lieb-Liniger model can be obtained using the Thermodynamic Bethe Ansatz (TBA) formalism it is also useful to have simple analytical formulae valid in different regimes which can be easily compared with experimental data.

The analytic results reported cover the six different regimes of the LL model (quasicondensate, nearly ideal Bose gas, and strongly interacting each of them with two subregimes) and are presented as expansions in the dimensionless strength γ and reduced temperature τ. Numerical comparisons with the TBA results highlighting the area of applicability of the various approximations and the crossover boundaries of the different regimes are also presented. The authors also use these results to investigate the density profiles in a trapped system using the local density approximation and propose a new method of thermometry for the LL model.

Probably the most interesting and important result is represented by the expansions derived for the thermal quasicondensate regime (regime II) which shows that the  temperature contributions to the pressure and pair correlation functions are not negligible as previously was believed (this also can be used to distinguish the I and II regimes). In addition, the newly derived expansions for regimes III (degenerate nearly ideal Bose gas regime) and IV (non-degenerate nearly ideal Bose gas regime) contain additional γ dependent terms compared with the results derived using the Hartree Fock approximation.

This is a clearly written and well-organized review that contains both previously known and new important original results reasons for which I recommend its publication in SciPost Physics Core.

Requested changes

None.

Recommendation

Publish (meets expectations and criteria for this Journal)

  • validity: high
  • significance: good
  • originality: good
  • clarity: top
  • formatting: excellent
  • grammar: excellent

Author:  Karen Kheruntsyan  on 2024-06-07  [id 4542]

(in reply to Report 1 on 2024-05-09)
Category:
remark

We are grateful to our Referee 1 for their careful reading of the manuscript and for providing valuable feedback. They have no specific questions or requests and suggest publication of our manuscript as is, and we are happy with this recommendation.

---

## Round 2 · Referee Report · Anonymous (Referee 2) · 2024-5-30

Report

In this work the authors present a thorough study of different regimes of the Lieb-Liniger model at finite temperature. The presented results summarise earlier works and contain new contributions. The main theme of the work is to identify different regimes and perform appropriate approximations to obtain analytical formulas capturing the essence of the physics in each regime but also include the subleading corrections to identify crossovers between the regimes. The approximate analytic results are then cross-checked with exact numerics.
The paper is well-written and nicely structured which helps in understanding the multitude of the results presented.

I think this work perfectly fits within the scope of the SciPost Physics Core. Still I have two comments to authors:

1) In the spirit of completing the program of characterising the regimes of the Lieb-Liniger model, I would like the authors to comment on the large T behaviour. There one could expect that quantum statistics stops to play any role and regardless of value of γ we approach a classical ideal gas. This is the case for the ideal 3d quantum gases and would be worthwhile to know if (and how) the restricted dimensionality influences that result.

2) I find a bit misleading the way the authors use the word LDA in Sec. 5. All of the computations presented in that section, as far as I understand, rely on the local density approximation. This includes the "exact TBA" computations. The analytic formulas for the density profile additionally combine the LDA with the approximate expressions for the thermodynamics. I would suggest rewording this section to clarify the situation.

Requested changes

As listed above

Recommendation

Ask for minor revision

  • validity: high
  • significance: good
  • originality: good
  • clarity: high
  • formatting: excellent
  • grammar: excellent

Author:  Karen Kheruntsyan  on 2024-06-07  [id 4543]

(in reply to Report 2 on 2024-05-30)
Category:
remark
answer to question

We thank Referee 2 for thoroughly engaging with our manuscript and providing such an extensive report. In particular, Referee 2 believes that "this work perfectly fits within the scope of the SciPost Physics Core", however, they make two comments, which we are pleased to address below.

Referee's comment 1:
In the spirit of completing the program of characterising the regimes of the Lieb-Liniger model, I would like the authors to comment on the large T behaviour. There one could expect that quantum statistics stops to play any role and regardless of value of 𝛾 we approach a classical ideal gas. This is the case for the ideal 3d quantum gases and would be worthwhile to know if (and how) the restricted dimensionality influences that result.

Our Reply:
The Referee is certainly right when saying that for any fixed small or large value of the interaction strength, determining the bosonic or fermionized behaviour, respectively, the system approaches the Maxwell-Boltzmann regime of the classical ideal gas at very high temperatures. In this sense, the 1D quantum gas is no different from its 3D counterparts, even though the regime of fermionization is absent in 3D.

This can be easily seen from the diagram of regimes in Fig. 1 (a), by tracing a vertical line at any fixed interaction strength. By increasing the temperature, the gas first enters regime IV corresponding to the non-degenerate nearly ideal Bose gas. By further raising the temperature, the system behaves as a classical ideal gas.

Furthermore, at very high temperatures, the perturbative terms of the interaction strength in the thermodynamic properties of regime IV in Sec. 4.2.2 become negligible compared to the leading temperature-dependent contributions of the ideal Bose gas (IBG), hence recovering the expected classical ideal gas result.

We have added a comment at the end of SubSec. 4.3.1 (pag. 24) in the manuscript, which explains this.

The changes in the manuscript diff.pdf file can be viewed here: https://www.dropbox.com/scl/fi/887wx4p1g06nd0vhsk3tq/diff.pdfrlkey=crsx2w3m2b71fsk54xvevm0nn&dl=0

Referee Comment 2):
I find a bit misleading the way the authors use the word LDA in Sec. 5. All of the computations presented in that section, as far as I understand, rely on the local density approximation. This includes the "exact TBA" computations. The analytic formulas for the density profile additionally combine the LDA with the approximate expressions for the thermodynamics. I would suggest rewording this section to clarify the situation.

Our Reply:
We believe that the Referee meant Sec. 4.4 here, rather than Sec. 5.

The Referee is right as, indeed, we generalized our results, previously derived in uniform configurations, to inhomogeneous trapped systems, by applying the LDA to:
i) exact TBA calculations;
ii) approximate analytic limits for the chemical potential in the six regimes, which allowed us to obtain the numerical spatial dependence of the density with an inversion procedure;
iii) by retaining only the first leading terms in the analytic limits for the chemical potential, we obtained analytic approximations for the entire density profiles which are valid only in certain regimes.

Results i)-iii) for the density profiles have been reported in Fig. 3.

The method for obtaining findings iii) was already discussed just above Eq. (138)
at pag. 27.

However, we reworded the 2nd paragraph on page 29 to emphasize the reliance on LDA in all cases.

The changes in the manuscript diff.pdf file can be viewed here:
https://www.dropbox.com/scl/fi/887wx4p1g06nd0vhsk3tq/diff.pdfrlkey=crsx2w3m2b71fsk54xvevm0nn&dl=0

---

## Editorial Decision

published